# Osteoprotegerin-dependent M cell self-regulation balances gut infection and immunity

Shunsuke Kimura[1,2,3,14]*, Yutaka Nakamura[1,14], Nobuhide Kobayashi[1], Katsuyuki Shiroguchi[3,4,5], Eiryo Kawakami[6], Mami Mutoh[7], Hiromi Takahashi-Iwanaga[2], Takahiro Yamada[1], Meri Hisamoto[8], Midori Nakamura[9], Nobuyuki Udagawa[9], Shintaro Sato[10,11], Tsuneyasu Kaisho[12], Toshihiko Iwanaga[2] & Koji Hase [1,13]*

Microfold cells (M cells) are responsible for antigen uptake to initiate immune responses in the gut-associated lymphoid tissue (GALT). Receptor activator of nuclear factor-κB ligand (RANKL) is essential for M cell differentiation. Follicle-associated epithelium (FAE) covers the GALT and is continuously exposed to RANKL from stromal cells underneath the FAE, yet only a subset of FAE cells undergoes differentiation into M cells. Here, we show that M cells express osteoprotegerin (OPG), a soluble inhibitor of RANKL, which suppresses the differentiation of adjacent FAE cells into M cells. Notably, OPG deficiency increases M cell number in the GALT and enhances commensal bacterium-specific immunoglobulin production, resulting in the amelioration of disease symptoms in mice with experimental colitis. By contrast, OPG-deficient mice are highly susceptible to *Salmonella* infection. Thus, OPG-dependent self-regulation of M cell differentiation is essential for the balance between the infectious risk and the ability to perform immunosurveillance at the mucosal surface.

---

[1] Division of Biochemistry, Faculty of Pharmacy and Graduate School of Pharmaceutical Science, Keio University, Tokyo 105-8512, Japan. [2] Laboratory of Histology and Cytology, Graduate School of Medicine, Hokkaido University, Sapporo 060-8638, Japan. [3] PRESTO, Japan Science and Technology Agency, Saitama 332-0012, Japan. [4] Laboratory for Prediction of Cell Systems Dynamics, RIKEN Center for Biosystems Dynamics Research (BDR), Suita 565-0874, Japan. [5] Laboratory for Immunogenetics, RIKEN Center for Integrative Medical Sciences (IMS), Yokohama 230-0045, Japan. [6] RIKEN Medical Sciences Innovation Hub Program (MIH), Yokohama 230-0045, Japan. [7] Department of Orthodontics, Faculty of Dental Medicine and Graduate School of Dental Medicine, Hokkaido University, Sapporo 060-8586, Japan. [8] Department of Oral Functional Prosthodontics, Division of Oral Functional Science, Graduate School of Dental Medicine, Hokkaido University, Sapporo 060-8586, Japan. [9] Department of Biochemistry, Matsumoto Dental University, Nagano 399-0781, Japan. [10] Mucosal Vaccine Project, BIKEN Innovative Vaccine Research Alliance Laboratories, Research Institute for Microbial Diseases, Osaka University, Osaka 565-0871, Japan. [11] Mucosal Vaccine Project, BIKEN Center for Innovative Vaccine Research and Development, The Research Foundation for Microbial Diseases of Osaka University, Osaka 565-0871, Japan. [12] Department of Immunology, Institute of Advanced Medicine, Wakayama Medical University, Wakayama 641-8509, Japan. [13] Division of Mucosal Barriology, International Research and Development Center for Mucosal Vaccines, The Institute of Medical Science, The University of Tokyo (IMSUT), Tokyo 108-8639, Japan. [14] These authors contributed equally: Shunsuke Kimura, Yutaka Nakamura. *email: kimura-sn@pha.keio.ac.jp; hase-kj@pha.keio.ac.jp

The mucosal surfaces of gastrointestinal tracts are exposed to ingested antigens and commensal microbes. The dome-shaped follicle-associated epithelium (FAE), specializing in luminal antigen uptake for immunosurveillance, is characterized by the presence of microfold (M) cells[1,2]. Antigen transcytosis across the intestinal epithelium via M cells is well documented to initiate mucosal immune responses. Glycoprotein 2 (GP2), expressed on the apical surface of M cells, serves as an uptake receptor for a subset of commensal and pathogenic bacteria[1]. GP2-deficient mice demonstrate attenuated T cell responses against orally infected Salmonella enterica serovar Typhimurium (S. Typhimurium), resulting in a diminished production of antigen-specific secretory IgA (SIgA)[1]. Furthermore, mice lacking M cells exhibit profound delays in mucosal immune system maturation, characterized by active germinal center (GC) reactions and IgA plasma cell development[3], as well as a reduction in antigen-specific T cell activation in Peyer's patches[4]. M cells are therefore considered to be beneficial for the onset of mucosal immune responses. Additionally, antigen transport by M cells may provide vulnerable gateways in the robust epithelial barrier. Indeed, several pathogenic bacteria, toxins, and a scrapie prion protein exploit M cells as entry portals to bypass the epithelial barrier and establish systemic infection[2]. These prior observations indicate that rigorous control of M-cell number within the intestinal epithelium is critical for maintenance of mucosal immunity.

M cells arise from intestinal stem cells via stimulation by the receptor activator of nuclear factor κB ligand (RANKL)[5,6]. Mice carrying an intestinal epithelium-specific deletion of RANK, a receptor for RANKL, lack M cells in the FAE of the gut-associated lymphoid tissue (GALT)[3]. Exogenous administration of RANKL in wild-type (WT) mice induces ectopic formation of M cells in the intestinal villi. RANKL belongs to the tumor necrosis factor cytokine superfamily, and is well documented to regulate osteoclast differentiation, as well as lymphoid organogenesis and development of γδT cell progenitors and medullary thymic epithelial cells[7–10]. RANKL binds to its receptor, RANK, to activate TRAF6 and its downstream non-canonical nuclear factor-κB (NF-κB) (p50/RelA) signaling pathways[9]. p50/RelA signaling subsequently upregulates expression of the non-canonical NF-κB molecule, p52/RelB[11], which transactivates two master transcription factors for M-cell differentiation, namely, Spib and Sox8 (refs. [4,6,12,13]). Newly generated Spi-B+Sox8+ M cells lack GP2 expression and exhibit an immature phenotype. These cells terminally differentiate into functionally mature Spi-B+ Sox8+ GP2high M cells during migration from the FAE-associated crypts into the dome region[13,14]. The RANK-RelB-Spi-B/Sox8 axis is responsible for differentiation and functional maturation into GP2high M cells. Stem/progenitor cells residing in the FAE-associated crypts are continuously exposed to RANKL from specialized stromal cells, known as M-cell inducer cells[15]. Nevertheless, a small portion (~10–20%) of all FAE cells ultimately become M cells. Furthermore, the number of GP2high mature M cells is reportedly significantly lower in the FAE of cecal patches than in the FAE of Peyer's patches[14]. These observations suggest the existence of suppression mechanisms of M-cell differentiation. However, the molecular machinery that regulates M-cell differentiation remains to be elucidated.

RANKL signaling is impeded by the binding of the soluble decoy receptor osteoprotegerin (OPG)[9,16,17], which negatively regulates osteoclast differentiation; thus, the RANKL–OPG balance is related to osseous diseases, including rheumatoid arthritis, osteoporosis, and periodontal disease. Interestingly, OPG is also known as a biomarker for inflammatory bowel diseases (IBD), namely, Crohn's disease and ulcerative colitis[18,19]; this suggests that an imbalance of RANKL–OPG may contribute to the pathogenesis of IBD by affecting gut immunity in a manner separate from its function in osteoimmunology.

Here, we propose a novel role for OPG in the self-regulatory machinery for the maintenance of M-cell density in the intestine. The absence of OPG increases the population of functionally mature M cells, thereby facilitating commensal-specific humoral immune responses in the GALT. This enhanced humoral response likely provides a protective barrier function against bacterial leakage from the gut lumen, given that the symptoms of experimentally induced colitis are alleviated in $Opg^{-/-}$ mice. In contrast, $Opg^{-/-}$ mice are highly susceptible to mucosal infection by pathogenic bacteria because of the augmentation of bacterial translocation via M cells. The exquisite balance of M-cell density found in the GALT may have been established over the course of evolution to maintain intestinal homeostasis in the context of invasion and immune responses of the mucosa.

## Results

**M cells secrete OPG into the subepithelial dome region.** To explore the molecular basis for the regulation of M-cell differentiation, we performed a high-resolution transcriptome analysis using digital RNA sequencing[20]. Although M cells constitute a very small proportion of the cellular population in the intestinal epithelium, the digital RNA sequencing technique enabled us to profile whole transcriptomics of as few as 100 cells. We confirmed that several known M-cell markers (e.g., Gp2, Spib, Tnfaip2, and Ccl9) were enriched in the gene cluster of GP2high M cells (Supplementary Data 1)[21]. A Kyoto Encyclopedia of Genes and Genomes (KEGG) pathway analysis revealed that two major pathways, namely cytokine–cytokine receptor interaction and osteoclast differentiation, were significantly activated in M cells (Fig. 1a). Notably, OPG-encoding Tnfrsf11b manifested the highest or third highest expression among the genes involved in these pathways (Fig. 1b). Quantitative polymerase chain reaction (PCR) analysis also confirmed that the expression level of OPG mRNA was 26.5 ± 2.6-fold (mean ± standard error) higher in the FAE than in the VE (Fig. 1c).

To identify OPG-expressing cells in the FAE, we performed whole-mount immunostaining of the FAE using the FAE-sheet (Fig. 1d)[13,14]. We used three M-cell markers: Tnfaip2, GP2, and Spi-B. Tnfaip2 is expressed in the early to middle stage of M-cell differentiation and GP2 is a marker for mature M cells that have a high uptake capacity[14]. Immunostaining with these M-cell markers further showed that both Tnfaip2- and GP2-positive M cells selectively expressed OPG (Fig. 1e). Furthermore, transmission electron microscopy revealed that OPG-expressing cells exhibited characteristic M-cell morphological features, namely, an irregular brush border and basolateral pockets embracing lymphocytes (Supplementary Fig. 1)[22]. Importantly, OPG immunoreactions were present in a subset of crypt cells that expressed Spi-B, an early transcription factor essential for M-cell differentiation (Fig. 1f)[4,6,12].

To gain further evidence for M-cell-specific OPG expression, recombinant RANKL tagged with glutathione-S-transferase (GST-RANKL) was administered to WT mice to induce ectopic differentiation of M cells in the VE[5]. The Tnfrsf11b/Opg expression level was prominently elevated in the VE within 12 h after RANKL administration (Fig. 1g). The upregulation of Opg occurred concomitantly with that of Spib, and was followed by upregulation of Gp2 (Fig. 1g). These data indicate that OPG is induced at an early stage, accompanied by Spi-B, during the M-cell differentiation process.

We subsequently investigated the OPG distribution in Peyer's patches by immunofluorescence analysis. Positive signals for OPG were widely detected in RANKL-positive reticular cells in the

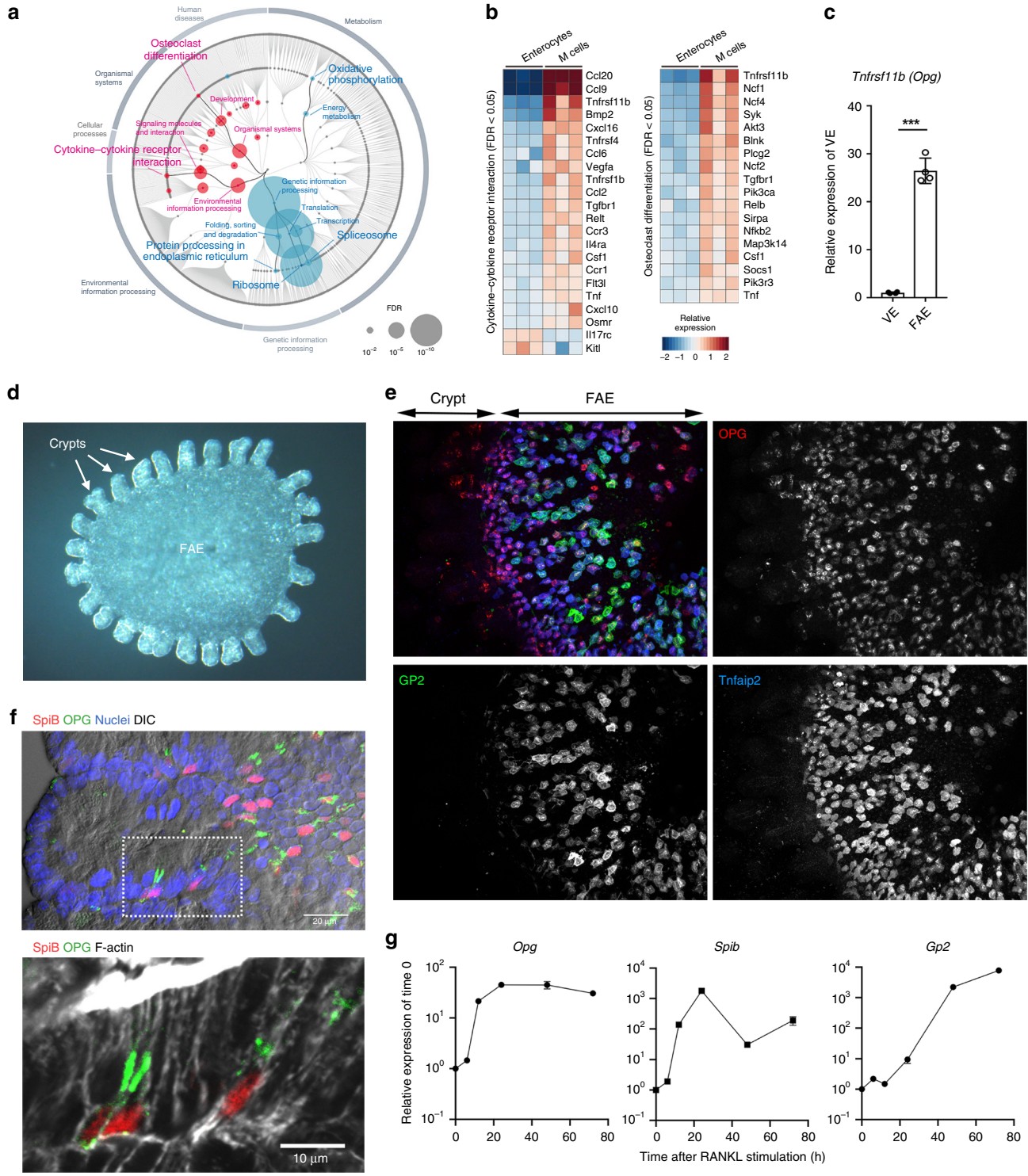

subepithelial dome (SED) region as well as in M cells in the FAE of Peyer's patches (Supplementary Fig. 2a). Because OPG is a secretory protein, immunofluorescence staining may detect positive signals in bona fide OPG producers, as well as surrounding cells that express RANKL. Indeed, fluorescent in situ hybridization analysis revealed that OPG mRNA was almost exclusively expressed by *Spib*-positive M cells, whereas it was rarely expressed in the SED region (Supplementary Fig. 2b). On the basis of these observations, we concluded that OPG released by M cells is distributed throughout the SED region of the GALT.

**OPG deficiency promotes M-cell differentiation**. Given the inhibitory effect of OPG on RANKL–RANK signaling, we hypothesized that OPG may play a role in the regulation of M-cell differentiation. We analyzed M cells in the GALT of $Opg^{-/-}$ mice by whole-mount immunostaining. Strikingly, we observed increased numbers of both Tnfaip2- and GP2-positive M cells in the Peyer's patches of $Opg^{-/-}$ mice (Fig. 2a, b, Supplementary Fig. 3a). Consistent with these results, several M-cell-associated genes were upregulated in the FAE of $Opg^{-/-}$ mice (Fig. 2c). Our scanning electron microscope observations revealed a prominent

**Fig. 1 M cells express osteoprotegerin from the early stage of differentiation. a** Enrichment analysis based on KEGG functional hierarchy for gene expression in M cells relative to their expression in enterocytes. Node size indicates the false-discovery rate of the parametric enrichment analysis. Red and blue nodes indicate respective significantly upregulated and downregulated pathways in M cells. **b** Gene expression profiles of enterocytes and M cells are shown. The heat map colors represent logFC for expression levels of genes compared with the mean expression value of each gene in enterocytes. **c** Increased expression of *Tnfrsf11b* (*Opg*) in the FAE of Peyer's patches, relative to expression in villus epithelium (VE). Results are normalized to *Gapdh* expression and are presented relative to the expression in the mean of VE. Values are presented as the mean ± standard error. ***$p < 0.005$ (Student's t-test, $n = 4$ animals). **d** Stereomicroscopic image of a whole-mount specimen of FAE monolayer and associated crypts, which was separated from the follicle by manipulation after treatment with EDTA as described in the Methods. The FAE area was surrounded by 12 crypts, as indicated by arrows. Scale bar: 200 μm. **e** OPG (red) is expressed in Tnfaip2 (blue)-positive and GP2 (green)-positive M cells in the FAE. Scale bars: 50 μm. **f** OPG (green) is expressed in Spi-B (red)-positive M cells of a FAE-associated crypt. The lower panel is an enlarged view of the area within the square in the upper panel. Cell nuclei were stained with DAPI (blue, left panel). F-actin was stained with phalloidin (gray, right panel). DIC differential interference contrast image. Scale bars: upper panel, 20 μm; lower panel, 10 μm. **g** *Opg* is an early expressing gene in the ileal epithelium after RANKL administration. Results were normalized to *Gapdh* expression and are presented relative to the expression in the epithelium without GST-RANKL treatment (time 0). Values are presented as the mean ± standard error. All data are representative of two (**c**) or three independent experiments (**a**, **b**, **d**, **e**–**g**). The source data underlying panels **c** and **g** are provided as a Source Data file.

increase in cells that had surface morphological features characteristic of M cells, within the FAE of $Opg^{-/-}$ mice, relative to their proportion in WT mice; this corroborated the augmentation of M-cell induction (Supplementary Fig. 3b). Immunofluorescence analysis manifested that the expression and localization of RANKL and RANK in the Peyer's patches of $Opg^{-/-}$ mice were similar to those observed in WT mice (Supplementary Fig. 4). M cells were previously reported to be inducible in epithelial organoids from intestinal crypts, upon stimulation with exogenous RANKL[5]. We confirmed that OPG deficiency facilitated the development of M cells in organoid cultures, as evidenced by the upregulation of both protein and mRNA expression of GP2 and Spi-B (Supplementary Fig. 5).

We previously reported that GP2[high] M cells displayed a functionally mature phenotype represented by the active uptake of luminal microbeads[14]. Whereas Peyer's patches harbor abundant mature M cells (Fig. 2a), we rarely observed mature M cells in the cecal patches of WT mice (Fig. 2d, e)[14]. In contrast, the cecal patches of $Opg^{-/-}$ mice harbored substantial numbers of mature M cells (Fig. 2d, e). The uptake of nanoparticles into cecal patches was augmented approximately 13-fold in $Opg^{-/-}$ mice, compared with the uptake in WT control mice (Fig. 2f and Supplementary Fig. 3c); however, the uptake of nanoparticles into Peyer's patches did not increase in $Opg^{-/-}$ mice (Fig. 2f). Nonetheless, these data strongly suggest that OPG functions as a negative regulator of M-cell differentiation in a cell-autonomous fashion.

**OPG[high] M cells are abundant in the cecal patches**. To investigate whether OPG is responsible for the suppression of M cell maturation in cecal patches, we compared OPG expression levels between Peyer's patches and cecal patches by image cytometry analysis (Fig. 3). We found that the expression level of OPG varied widely among M cells, and that the number of OPG[high] M cells was significantly higher in cecal patches than in Peyer's patches.

Furthermore, we quantified the proportion of Spi-B-positive cells among OPG-positive cells by image cytometry analysis (Fig. 3b, c). The results demonstrated that 98.6% ± 1.3% (mean ± standard deviation) of OPG-positive cells were Spi-B-positive in Peyer's patches and 94.6% ± 3.7% of OPG-positive cells were Spi-B-positive in cecal patches. These quantitative data support our observations that the major OPG-producing cells were M cells.

**OPG deficiency enhances RANKL signaling in the gut epithelia**. We subsequently investigated the effect of OPG on M-cell induction in the ileal and cecal epithelium. As we previously

reported, exogenous RANKL treatment induced differentiation of GP2[high] mature M cells in the ileal, but not cecal, epithelium (Fig. 4a)[14]. The expression level of RANK mRNA in the cecal epithelium was similar with that in the ileal epithelium (Supplementary Fig. 6). Surprisingly, RANKL administration to $Opg^{-/-}$ mice promoted differentiation of GP2[high] M cells in the cecal epithelium (Fig. 4a). Quantitative PCR analysis also confirmed a remarkable (approximately $2 \times 10^4$-fold) upregulation of *Gp2* in the cecal epithelium of $Opg^{-/-}$ mice upon treatment with RANKL, whereas the *Gp2* expression level in this region of WT mice was unchanged by RANKL treatment (Fig. 4b). Nevertheless, the expression of *Spib* and *Tnfaip2*, the early-to-middle M-cell differentiation markers, were upregulated even in the cecal epithelium of WT mice upon RANKL injection (Fig. 4b). Thus, OPG seems to suppress RANKL–RANK signaling mainly at late stage of M-cell differentiation.

To clarify the influence of OPG on regulatory mechanisms of M-cell differentiation, we focused on the activation of RelB, a member of the NF-κB family. RelB is retained in the cytoplasm in its latent inactive form; upon activation, it is rapidly transported into the nucleus. This event is indispensable for the initiation of M-cell differentiation and maturation[11,14]. Immunoblot analysis revealed that, after RANKL administration, the amounts of nuclear RelB in both ileal and cecal epithelium were higher for $Opg^{-/-}$ mice than for WT mice (Fig. 4c). Consistent with this, the proportion of Spi-B-positive M cells harboring nuclear RelB was significantly greater in the cecal FAE of $Opg^{-/-}$ mice than in that of WT mice (Fig. 4d). These data imply that OPG suppresses RelB activity in M cells, likely through reduction of the intensity or duration of RANKL–RANK signaling; this mechanism aids in maintenance of appropriate M-cell density.

**Absence of OPG promotes immune responses in the gut**. We further explored the biological significance of OPG-dependent self-regulation of M-cell density in the mucosal immune system. GALT continuously undergoes GC reactions, including class switch recombination and somatic hypermutation, to generate high-affinity IgA[+] cells[23]. To determine whether increases in M-cell number and M-cell-mediated particulate antigen uptake affected the immune response, we analyzed immune cell subsets in the ileal Peyer's patches, cecal patches, and colons of $Opg^{-/-}$ mice. Total B cells were expanded in the cecal patches and the colon, but not in the ileal Peyer's patches, of $Opg^{-/-}$ mice (Fig. 5a and Supplementary Fig. 7a). Accordingly, the number, but not frequency, of GC B cells increased in the cecal patches and colons embracing solitary intestinal lymphoid tissues[24] (Fig. 5b and Supplementary Fig. 7b). This observation was consistent with the finding that expansion of the M-cell population was more

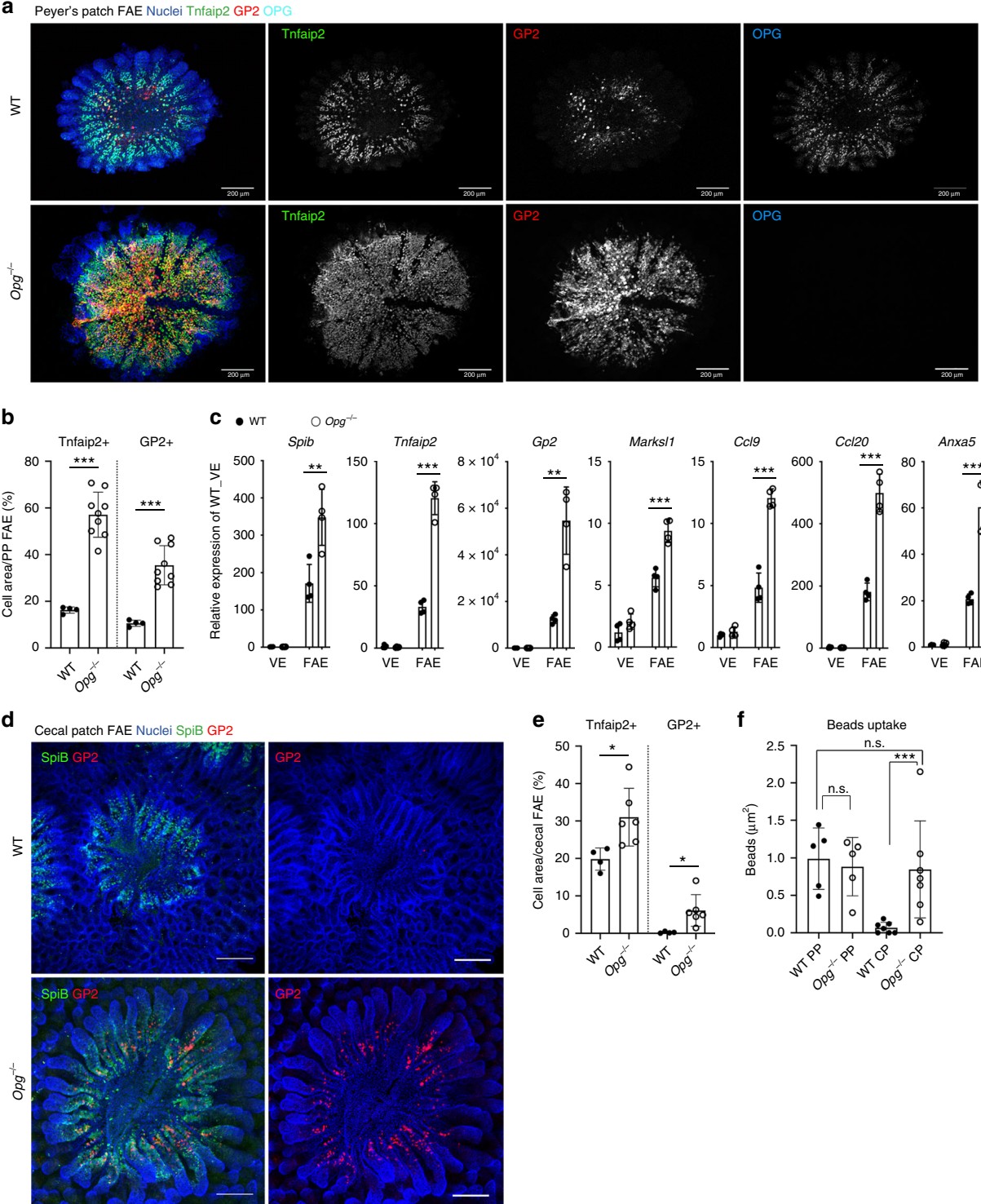

**Fig. 2 M-cell differentiation and activation are enhanced in the absence of OPG. a** The numbers of M cells labeled with Tnfaip2 (green) and GP2 (red) are prominently increased in the follicle-associated epithelium (FAE) of Payer's patches from $Opg^{-/-}$ mice. Each indicated channel is shown separately in gray scale. Scale bars: 200 μm. **b** The numbers of Tnfaip2-positive or GP2-positive M cells in the FAE of Peyer's patches from WT and $Opg^{-/-}$ mice were quantified. Values are presented as the mean ± standard deviation. ***$p < 0.005$ (Student's $t$-test, $n = 4$ of WT and 9 of $Opg^{-/-}$ mice). **c** The expression levels of M cell-associated genes in the FAE of Peyer's patches were normalized to $Gapdh$ expression and are presented relative to the expression in the villus epithelium (VE). Values are presented as the mean ± standard deviation. **$p < 0.01$, ***$p < 0.005$ (Student's $t$-test, $n = 4$ animals). **d** Whole-mount staining images of GP2 (red), Spi-B (green), and DAPI (blue) in cecal FAE. Scale bars: 100 μm. **e** The numbers of Tnfaip2-positive or GP2-positive M cells in the FAE of cecal patches from WT and $Opg^{-/-}$ mice were quantified. Values are presented as the mean ± standard deviation. *$p < 0.05$ (Student's t-test, $n = 4$ of WT and 9 of $Opg^{-/-}$ mice). **f** The uptake of latex beads into Peyer's patches and cecal patches was measured. Data from three independent experiments are presented as the mean ± standard deviation. ***$p < 0.005$. n.s. not significant (Student's $t$-test, $n = 7$ animals). The source data underlying panels **b**, **c**, **e** and **f** are provided as a Source Data file.

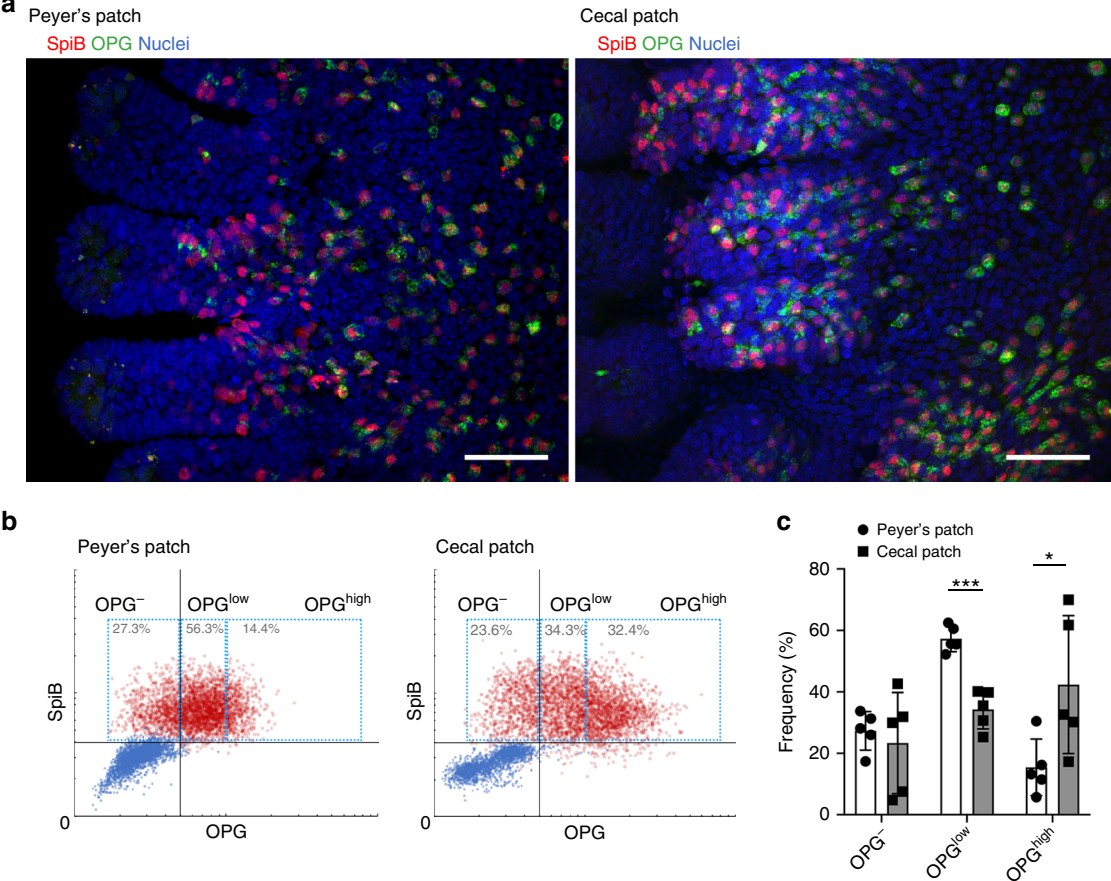

**Fig. 3 OPG^high M cells cluster more in cecal patches than in Peyer's patches. a** Whole-mount immunostaining of the FAE of Peyer's patches (left) and cecal patches (right) for OPG (green) and Spi-B (red). Nuclei were stained with DAPI (blue); scale bars: 50 μm. **b** Scatter plots of the fluorescence intensities of OPG versus Spi-B. Red dots represent cells stained with anti-Spi-B antibody that was conjugated with HyLyte Fluor 555 and anti-OPG antibody that was conjugated with HyLyte Fluor 647. Blue dots represent background fluorescence intensity of randomly selected non-stained cells. Fluorescence intensities were measured for at least 3000 cells from five FAEs of three mice. **c** Frequencies of OPG^high M cells in Peyer's patches and cecal patches were quantified. *$p < 0.05$, ***$p < 0.005$. Student's *t*-test, $n = 5$ FAE from three animals. The source data underlying panels **b** and **c** are provided as a Source Data file.

obvious in the distal GALT (Fig. 2c, d). The activation of GC reactions eventually increased the number of IgA class-switched B cells in the cecal patches and colons (Fig. 5c and Supplementary Fig. 7c), indicating that deletion of *Opg* facilitated mucosal humoral immunity in the distal GALT. Likewise, IgA^+ plasma cells was expanded in the colons of these mice (Fig. 5d and Supplementary Fig. 7e). Importantly, the number of IgG class-switched B cells also increased in the cecal patches and colons of *Opg*^−/− mice (Fig. 5c and Supplementary Fig. 7d), whereas IgG^+ plasma cells were nearly absent in the colon (Fig. 5d and Supplementary Fig. 7e). We also observed that the total number of CD4^+ T helper (Th) cells was significantly increased in the colon but not in the GALT, whereas the number of CD8^+ T cells remained unchanged in all tissues tested for *Opg*^−/− mice (Fig. 5e).

To further examine whether antigen-specific immunoglobulin production was affected in *Opg*^−/− mice, we employed a well-established oral immunization model using *S.* Typhimurium expressing fragment C of the tetanus toxoid (*Salmonella*-ToxC), which is taken up by M cells and induces a tetanus toxoid (TT)-specific mucosal immune response in GALT[1,25]. This strain was constructed from an attenuated Δ*aroA* Δ*aroD* parent strain, which possesses a type III secretion system but cannot multiply within host cells[25]. After oral immunization, the level of

TT-specific fecal SIgA was approximately three-fold higher in *Opg*^−/− mice than in control mice (Fig. 5f). The level of TT-specific IgG tended to increase in *Opg*^−/− mice, although the difference was not statistically significant. These results indicated that the absence of OPG consolidates the IgA and IgG responses to luminal antigens. In contrast, the IgG response to systemic immunization was slightly but significantly lower in *Opg*^−/− mice (Fig. 5g). This result implies that OPG, most likely expressed by certain immune cell subsets (Supplementary Fig. 8), may play a role in the induction of systemic immune response.

**Loss of OPG improves disease symptoms in DSS colitis.** The earliest visible lesions in recurrent Crohn's disease are microscopic erosions at the FAE[26], and patients with longstanding Crohn's disease exhibit increased uptake of non-pathogenic *Escherichia coli* via the FAE[27]. Because increased M-cell density in *Opg*^−/− mice may attenuate mucosal barrier function, we sought to determine whether the absence of *Opg* altered susceptibility to dextran sulfate sodium (DSS)-induced colitis. Unexpectedly, the symptoms of DSS-induced colitis were ameliorated in *Opg*^−/− mice, as evidenced by prevention of body weight loss and colonic shortening, lower levels of histological damage, and lower diarrhea severity, compared with co-housed or littermate control

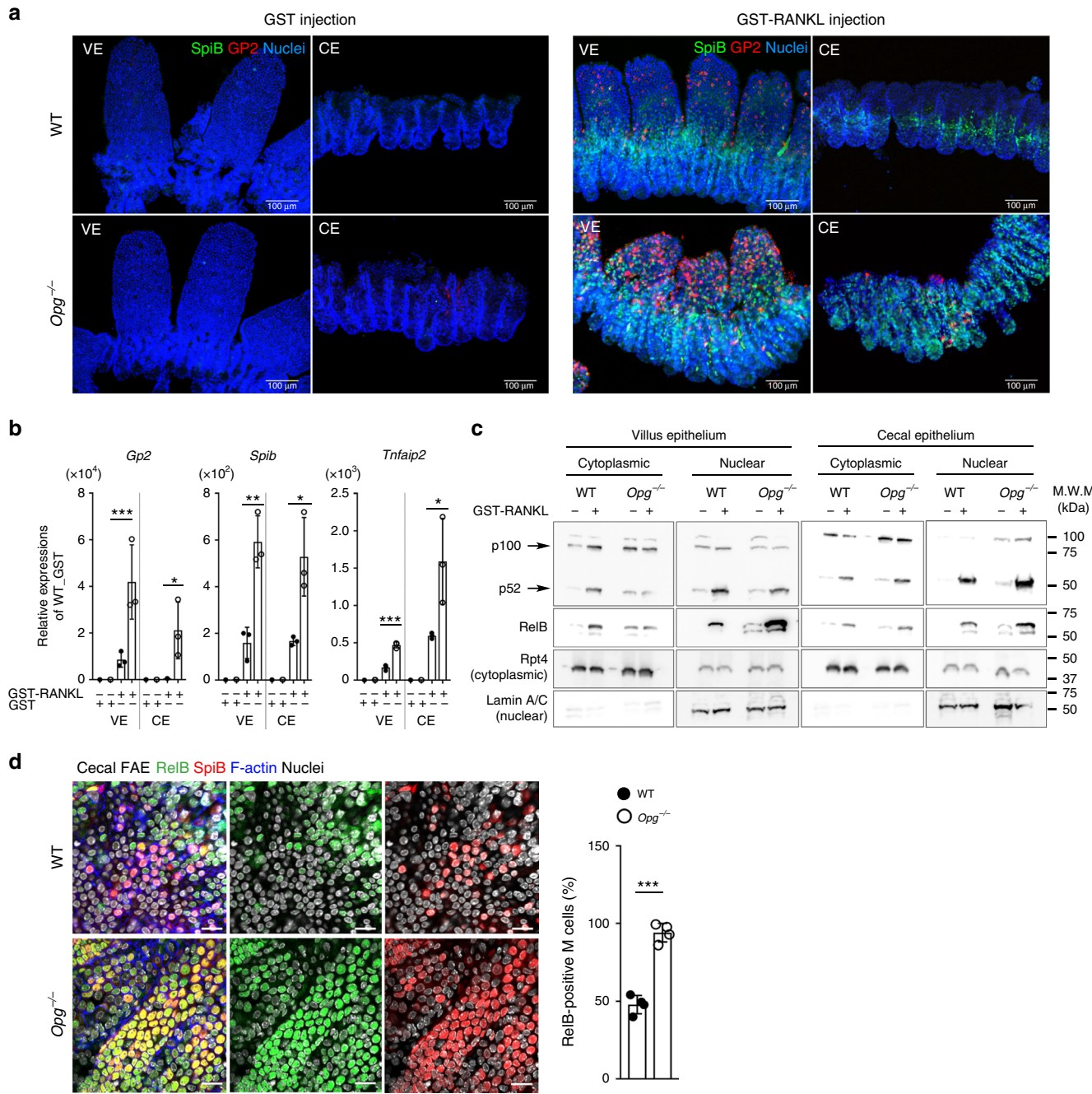

**Fig. 4 RANKL–RANK signaling is stimulated in the gut epithelia of Opg$^{-/-}$ mice. a** GP2$^+$ cells (red) are more effectively induced in the cecal epithelium (CE) and ileal villi (VE) of Opg$^{-/-}$ mice by RANKL administration. Whole-mount immunohistochemical images of Spi-B (green) and GP2 (red) in the VE and CE of WT (upper panels) and Opg$^{-/-}$ mice (lower panels) treated with either GST (control; left) or GST-RANKL (right). Scale bars: 100 μm. Representative images from three independent experiment are shown. **b** Quantitative PCR analysis of M-cell marker expression in conventional epithelia from the VE and CE of mice injected with GST (control) or GST-RANKL. Results were normalized to *Gapdh* expression and are presented relative to the expression in the ileal epithelium from GST-treated mice. Data shown are mean values from three independent experiments (error bars indicate standard deviation). ***$p < 0.005$, **$p < 0.01$, *$p < 0.05$; $p$ values were calculated with the Student's $t$-test ($n = 3$ biologically independent experiments). **c, d** Nuclear translocation activities of RelB and p52 following RANKL stimulation are enhanced in Opg$^{-/-}$ mice. **c** Western blot analysis of p100/p52 and RelB in the VE and CE of mice injected with GST or GST-RANKL. Rpt4, a subunit of the 26S proteasome, was used as an internal control for the cytoplasmic fraction. Lamin A/C (Lamin) was used as an internal control for the nuclear fraction. Data are representative of two independent experiments. **d** Right, single confocal planes of the cecal FAE from WT and Opg$^{-/-}$ mice. FAE monolayers were stained with anti-RelB (green) and anti-Spi-B (red) antibodies and with Hoechst 33342. Left, a bar graph summarizing the proportions of RelB-positive cells among total numbers of M cells (Spi-B-positive cells). ***$p < 0.005$; $p$ values were calculated with the Student's $t$-test ($n = 4$ biologically independent experiments). Scale bars: 20 μm. The source data underlying panels **b** and **d** and non-cropped scan images of western blotting (**c**) are provided as a Source Data file.

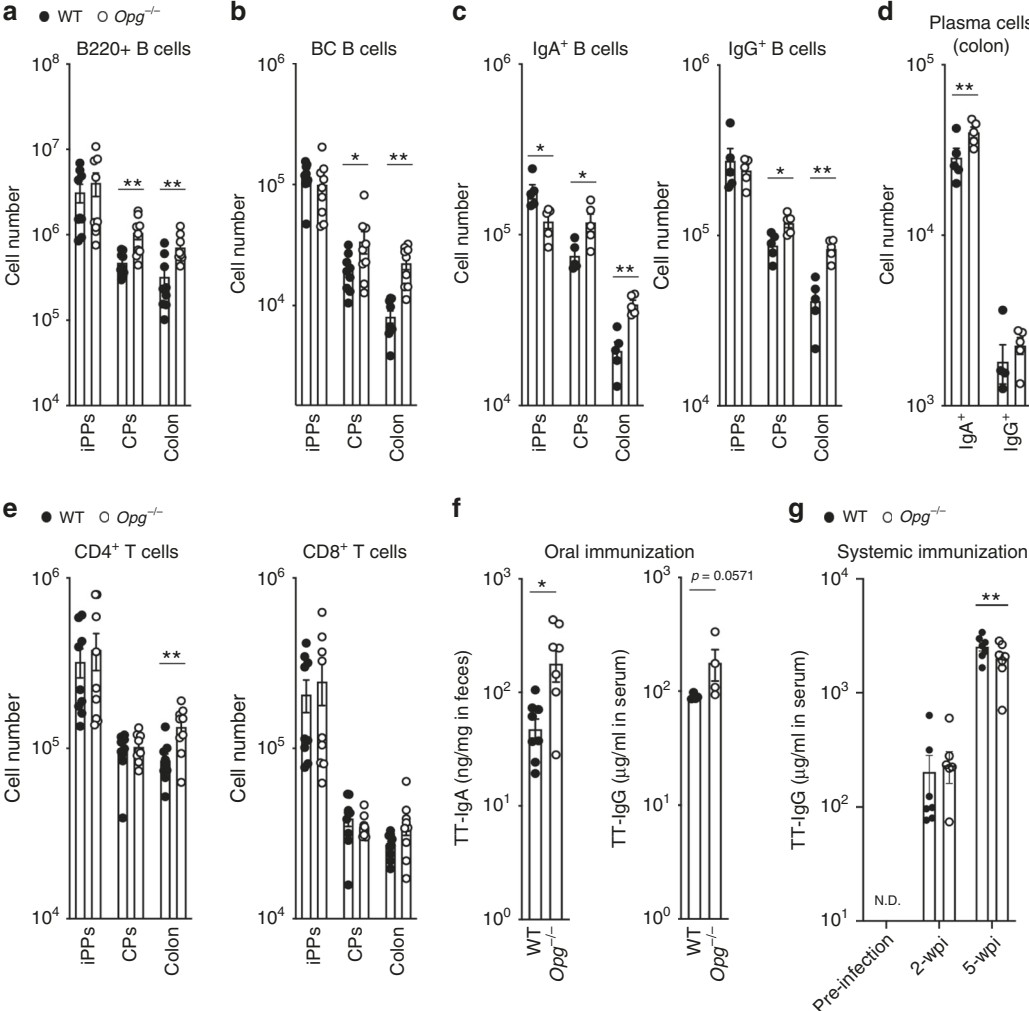

**Fig. 5 Absence of Opg promotes mucosal immune responses in the gut. a–c** The numbers of B cells (**a**), germinal center (GC) B cells (**b**), IgA+ B cells and IgG+ B cells (**c**) in ileal Peyer's patches (iPPs), cecal patches (CPs), and whole colons were quantified by flow cytometry. **d** The numbers of IgA+ and IgG+ plasma cells in the colons were quantified by flow cytometry. Data are presented as the mean ± standard error (**a**, **b**, **e**) or standard deviation (**c**, **d**). **\*\*p <** 0.01, **\*p < 0.05;** p values were calculated with the Student's t-test (**a** and **b**, n = 9 animals from two independent experiments; **c** and **d**, n = 5 animals, representative of two independent experiments). **e** The numbers of CD4+ T cells (left) and CD8+ T cells (right) in iPPs, CPs, and whole colon were quantified by flow cytometry. **\*\*p < 0.01;** Student's t-test, n = 9 animals from two independent experiments. **f** Feces and sera were collected from Opg−/− or co-housed WT mice at 4 weeks after oral immunization with 5 × 10⁷ c.f.u. of Salmonella-ToxC. The amounts of ToxC-specific-IgA in feces (left) and -IgG in serum (right) were evaluated by ELISA. Data are presented as the mean ± standard error (left) or standard deviation (right) (left, Welch's t-test; n = 9 animals from two independent experiments; right, Mann–Whitney U test, n = 4 animals of representative of two independent experiments). **g** Sera were collected from Opg−/− or littermate WT mice at indicated time periods after intraperitoneal immunization with 5 × 10⁵ c.f.u. of Salmonella-ToxC. The amount of ToxC-specific-IgG in serum was evaluated by ELISA. Data are presented as the mean ± standard error (n = 7 of littermate and co-housed WT, n = 7 of Opg−/− mice from two independent experiments). **\*p < 0.05** calculated with two-way ANOVA. The source data underlying panels **a–g** are provided as a Source Data file.

mice (Fig. 6a–d). DSS damages colonic epithelium and subsequently promotes systemic bacterial translocation, thereby causing splenomegaly[28]. Compared with control mice, DSS-induced splenomegaly was alleviated in Opg−/− mice (Fig. 6e). We further analyzed myeloid cell subsets that contribute to the development and regulation of colitis. At day 7 after the onset of DSS administration, neutrophil infiltration into the colon, which mediates inflammatory reactions, was significantly reduced in Opg−/− mice (Fig. 6f). Conversely, the numbers of Ly6C− resident macrophages and CD103+ dendritic cells were significantly increased in the colons of Opg−/− mice (Fig. 6f). These cell populations exhibit anti-inflammatory phenotypes[29,30]. Taken

together, the above observations illustrate that the absence of OPG limits the development of local and systemic inflammation.

Because OPG is expressed by hematopoietic cell lineages, such as B cells and dendritic cells, hematopoietic OPG may affect susceptibility to DSS-induced colitis. To test this hypothesis, we performed bone marrow-chimera experiments. We observed that the mice lacking OPG in non-hematopoietic, but not hematopoietic, cells recapitulated the phenotype of Opg−/− mice (Fig. 7). Because OPG is predominantly expressed by M cells in non-hematopoietic cell populations of the intestine, we concluded that the absence of M-cell-intrinsic OPG is responsible for the amelioration of colitis in Opg−/− mice.

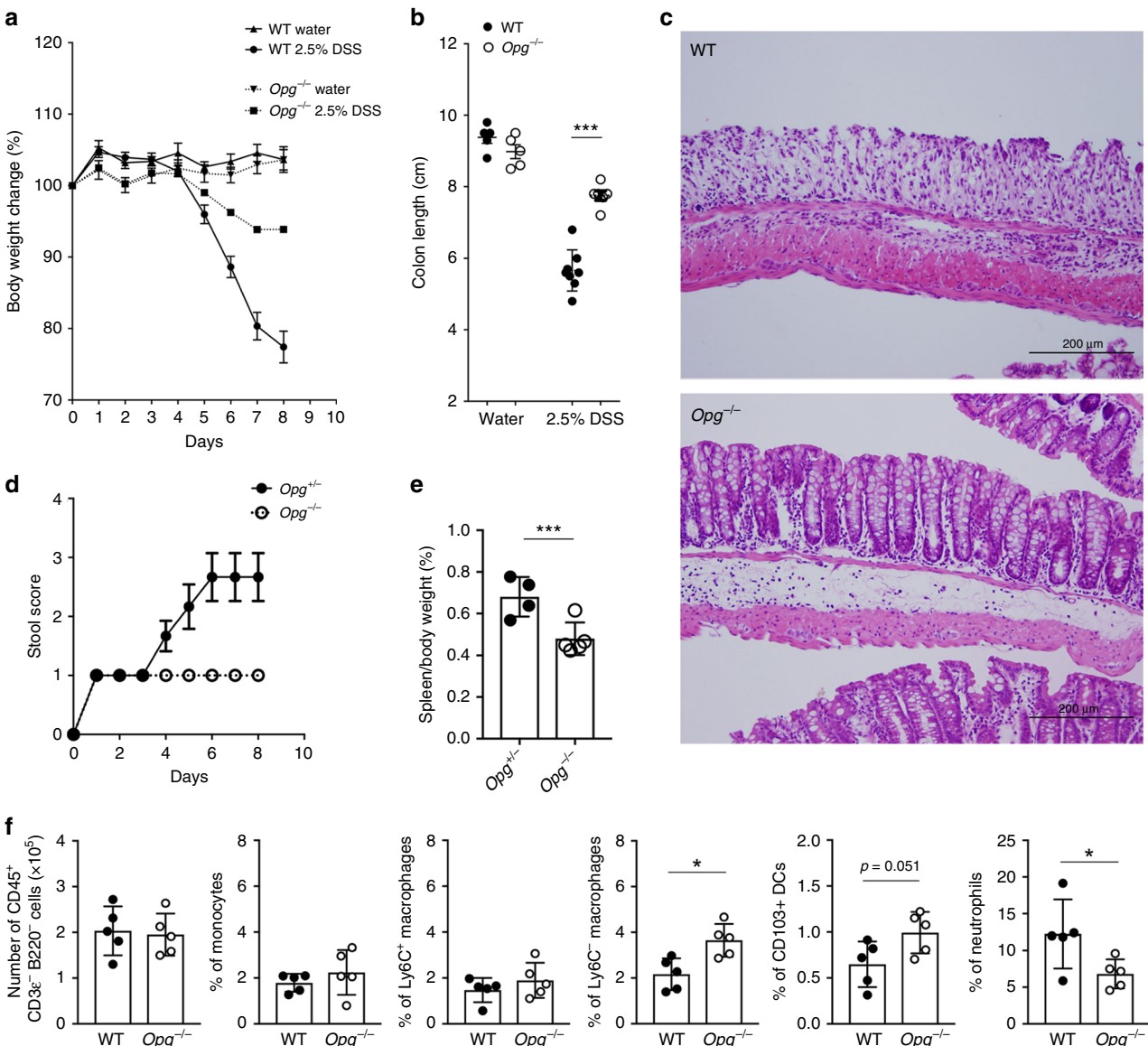

**Fig. 6 Absence of *Opg* ameliorates DSS-induced colitis. a** Daily changes in body weight during dextran sodium sulfate (DSS)-induced colitis. Changes in body weight percentage were calculated by dividing the body weight on the specified day by the body weight at day 0. **b** Colon length was measured after sacrifice at day 8. **c** Representative histology of hematoxylin- and eosin-stained colonic tissue from WT and *Opg*$^{-/-}$ mice with DSS-induced colitis on day 8. **d** Stool scores were measured as described in the Methods. **e** Spleen weight was measured after sacrifice at day 8. **a**, **b**, **d**, and **e** Data are presented as the mean ± SEM. **$p < 0.01$, ***$p < 0.005$ (Student's *t*-test, $n = 5$ animals). Similar results were obtained from three independent experiments, and representative data are shown. **f** The number of CD45$^+$CD3ε$^-$B220$^-$ cells and the frequency of monocytes, Ly6C$^+$ macrophages, Ly6C$^-$ macrophages, CD103$^+$ dendritic cells (DCs), and neutrophils in the population of CD45$^+$CD3ε$^-$B220$^-$ cells of the colonic lamina propria were quantified by flow cytometry. Data are representative of three independent experiments and are presented as the mean ± standard deviation. *$p < 0.05$; Student's *t*-test, $n = 5$ animals. The source data underlying panels **a**, **b**, **d**, **e** and **f** are provided as a Source Data file.

We further assessed the intestinal microbiota, which has a significant impact on susceptibility to intestinal inflammation[31], of WT and *Opg*$^{-/-}$ mice. There was no overt change in microbial composition between the two groups in the steady state (Supplementary Fig. 9a, b). Furthermore, species richness, as determined using the Chao1 index, and beta diversity were also unchanged in *Opg*$^{-/-}$ mice (Supplementary Fig. 9c, d). These data indicate that the reduced susceptibility to DSS-induced colitis in *Opg*$^{-/-}$ mice is not related to alteration of the microbial community.

In the steady state, the number of CD4$^+$ T cells was increased in the colonic lamina propria (cLP) of *Opg*$^{-/-}$ mice (Fig. 5e). CD4$^+$ T cells are known as the main drivers of IBD when this

balance is perturbed[32]. We further analyzed T cell subsets in the cLP of *Opg*$^{-/-}$ mice during DSS colitis; however, there were no significant differences in the numbers of Th1, Th17, and Treg cells (Supplementary Fig. 10).

**Loss of Opg reinforces the IgG response to gut microbiota.** Immunoglobulins (Igs) specific for commensal bacteria have been detected in the feces and sera of mice under physiological conditions[33,34]. The observed increases in IgA$^+$ and IgG$^+$ B cells in the distal GALT of *Opg*$^{-/-}$ mice (Fig. 5c) suggested that commensal-specific Igs may be enhanced in the absence of OPG, eventually restricting bacterial translocation and mitigating the

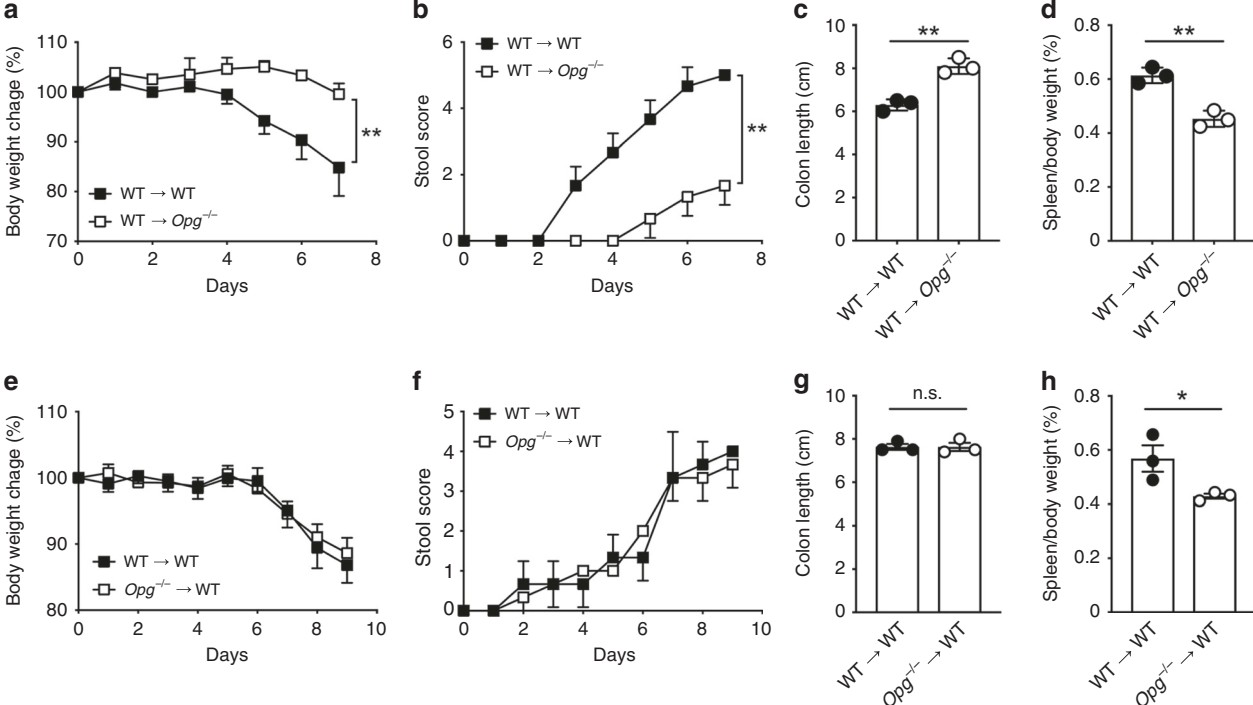

**Fig. 7 Non-hematopoietic cells in Opg$^{-/-}$ mice contribute to relief of DSS colitis symptoms. a, e** Daily changes in body weight during dextran sodium sulfate (DSS)-induced colitis. Changes in body weight percentage were calculated by dividing the body weight on the specified day by the body weight at day 0. **b, f** Fecal clinical scores were measured as described in the Methods. **c, g** Colon length was measured after sacrifice at day 8. **d, h** Spleen weight was measured after sacrifice at day 8. WT or Opg$^{-/-}$ recipient mice following bone marrow transplantation from WT donors (**a–d**) and WT recipient mice following bone marrow transplantation from WT or Opg$^{-/-}$ donors (**e–h**) treated with 1.5% DSS for 7 days. Data are representative of two independent experiments. **$p < 0.01$, n.s. not significant, $p$ values were calculated using unpaired $t$-test ($n = 3$ mice). The source data underlying panels **a–h** are provided as a Source Data file.

inflammatory response. To test this possibility, we first analyzed the association of Igs with commensal bacteria by flow cytometry (Fig. 8a). As anticipated, commensal bacteria were predominantly coated by IgA, regardless of whether the mice were subjected to physiological or inflammatory conditions (Fig. 8b, c). There was no significant difference in the frequency of IgA-coated bacteria between Opg$^{-/-}$ and WT mice (Fig. 8b). In contrast, this frequency was significantly increased in Opg$^{-/-}$ mice after DSS treatment (Fig. 8c).

We next attempted to detect serum Igs that recognized commensal bacteria. For this, we incubated each serum sample with fecal bacteria prepared from the corresponding mouse (Fig. 8d)[33]. The frequencies of IgA-, IgG1-, and IgG3-bound bacteria were significantly higher for serum from Opg$^{-/-}$ mice, compared with serum from co-housed WT mice under physiological conditions (Fig. 8e). This trend persisted under inflammatory conditions (Fig. 8f). Our results suggest that serum commensal-specific IgG, especially IgG3, is induced in an M-cell-dependent manner.

**IgG3$^+$ and IgA$^+$ cells play protective roles in DSS colitis.** A previous report demonstrated that luminal IgA and serum IgG prevent the development of DSS-induced colitis by neutralization of systemically translocated commensal bacteria[33]. We therefore hypothesized that protection against DSS-induced colitis in the absence of OPG results from augmented commensal-specific IgA and IgG3 responses. To test our hypothesis, we performed adoptive transfer of IgG3$^+$ or IgA$^+$ B cells into Rag1$^{-/-}$ recipient mice. Although the transfer of these B cell populations did not affect body weight loss, it led to significant reduction of colitis symptoms, such as fecal occult blood, diarrhea, colon shortening,

and colonic tissue hyperplasia (Supplementary Fig. 11). On the other hand, there is no significant difference in the protective effect of the B cell subsets between WT and Opg$^{-/-}$ mice (Supplementary Fig. 12).

**OPG reduces the invasion of M-cell-targeting pathogens.** Our findings reveal that unexpected advantages for mucosal immune responses result from enhanced development of M cells. However, this raised a fundamental question: why is self-regulation of M-cell differentiation required? Given that M cells serve as an entry portal for systemic invasion by certain pathogenic agents, such as S. Typhimurium[35], we hypothesized that OPG may be instrumental in lowering the risk of infection by M-cell-targeting pathogens. This hypothesis was tested using a mucosal infection model based on an S. Typhimurium aroA-deficient strain (ΔaroA), which can invade the host body via M cells; however, this attenuated strain cannot multiply within host cells[36]. The number of S. Typhimurium ΔaroA internalized into Peyer's patches was much higher in Opg$^{-/-}$ mice than in WT mice (Fig. 9a). To further confirm that the enhanced invasion in Opg$^{-/-}$ mice was attributed to the expansion of M cells, we orally infected the mice with Salmonella pathogenicity island 1 (SPI-1) mutant strain of S. Typhimurium, which cannot gain entry into M cells[37]. There was no significant difference in the number of the SPI-1 mutant bacteria translocated into the mesenteric lymph nodes between Opg$^{-/-}$ and WT mice (Supplementary Fig. 13). Thus, the uptake of M-cell-independent pathogens remains intact in the absence of OPG.

The enhanced translocation of the ΔaroA strain suggests that Opg$^{-/-}$ mice may be vulnerable to salmonellosis. We therefore infected Opg$^{-/-}$ and WT mice with lethal doses of toxic S. Typhimurium (Fig. 9b). Opg$^{-/-}$ mice succumbed to infection

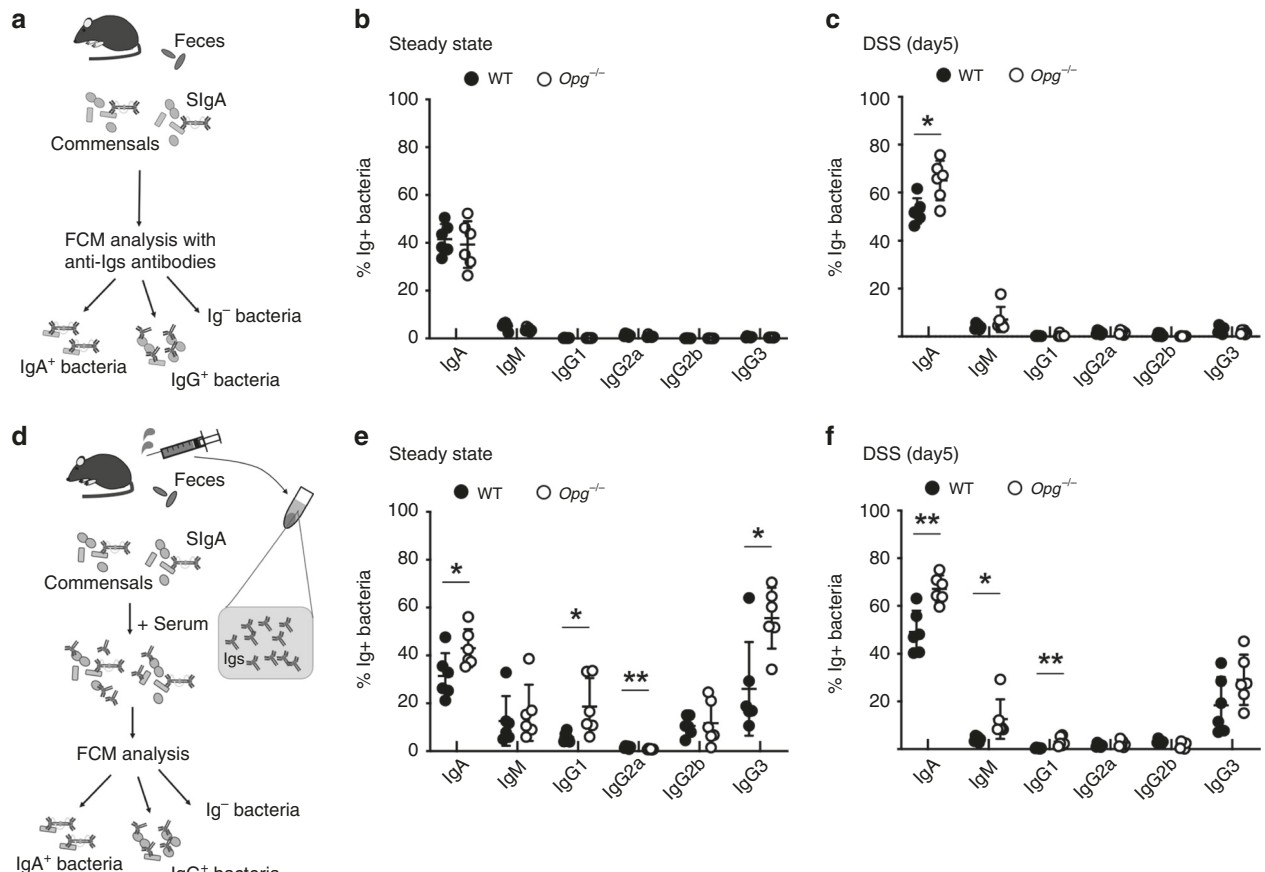

**Fig. 8 Commensal-specific antibody production is increased in $Opg^{-/-}$ mice. a, d** Schematic cartoons depicting experimental designs. Commensal-specific fecal or serum immunoglobulins were detected by flow cytometry. **b, c, e, f** Feces and sera were collected from $Opg^{-/-}$ or co-housed WT mice at day 0 (**b, e**) or day 5 (**c, f**) after the onset of DSS consumption. Flow cytometry assays of DAPI-positive fecal bacteria bound by the indicated Igs were performed without (**b, c**) or with (**e, f**) serum. Representative data from two independent experiments are shown as the mean ± standard deviation. *$p <$ 0.05, **$p <$ 0.01 (one-way ANOVA, $n = 6$ animals). The source data underlying panels **b, c, e** and **f** are provided as a Source Data file.

with *S.* Typhimurium significantly more rapidly than co-housed WT mice (Fig. 9c). Moreover, intraperitoneal RANKL treatment dramatically aggravated the infectious disease, as manifested by the shortened survival time of RANKL-treated $Opg^{-/-}$ mice (Fig. 9b, d). In this experiment, mice were infected with toxic *S.* Typhimurium just after the second administration of RANKL (Day 0). At this time, GP2 expression was induced in the intestinal villi of $Opg^{-/-}$ mice but not in WT mice (Supplementary Fig. 14). As the result, nearly half of the RANKL-treated $Opg^{-/-}$ mice died within 2 days post-infection, and all of these mice succumbed to lethal infection within 4 days; in contrast, WT mice survived until 8 days post-infection. These findings indicate that the regulation of M-cell development is critical for restriction of host invasion by M-cell-targeting pathogens.

## Discussion

We previously defined GP2 as a hallmark of M-cell full maturation: Spi-B+ GP2high cells are mature M cells that vigorously take up luminal antigens, whereas Spi-B+ GP2−/low cells are immature M cells with lower capacity to take up luminal antigens[13,14]. GP2high mature M cells were abundant in Peyer's patches but sparse in cecal patches, consistent with the reduced activation of RelB signaling. We have here shown that OPG is highly expressed by cecal M cells, which suppress the input of RANKL signals into surrounding cells and consequently reduce the activity of RelB. This strong self-repression of M cells by OPG

may repress the number of GP2-positive mature M cells, particularly within the cecal FAE. Genetic deletion of *Opg* resulted in prominent expansion of GP2high mature M cells in the FAE of cecal patches, with a higher level of nuclear RelB, as well as in the FAE of Peyer's patches. Due to this change, antigen-uptake efficiency significantly increased in cecal patches, whereas minimal change was observed in Peyer's patches. Notably, Peyer's patches exhibit large numbers of GP2high mature M cells when OPG is present, and M-cell-dependent antigen uptake may therefore be constitutively activated, such that it reaches a plateau. In contrast, highly upregulated OPG hinders the development of GP2high mature M cells in cecal patches. Eventually, incorporation of nanoparticles into cecal patches was nearly eliminated in WT mice, whereas OPG deficiency markedly enhanced the uptake of nanoparticles in association with the expansion of GP2high mature M cells. Consistent with this finding, mucosal immune responses, such as the GC reaction that leads to IgA and IgG class switching, were enhanced in cecal patches and the cLP in $Opg^{-/-}$ mice. These data imply that OPG-dependent control of M-cell development plays a critical role in the regulation of immune response, particularly in the distal GALT.

SIgA is the most abundant class of antibodies found on the mucosal surface; these antibodies play important roles in protection against pathogens and in the regulation of the gut microbial community[38]. We previously reported that the IgA response is gradually enhanced during the development of DSS-induced colitis[39]. Here, we found that commensal

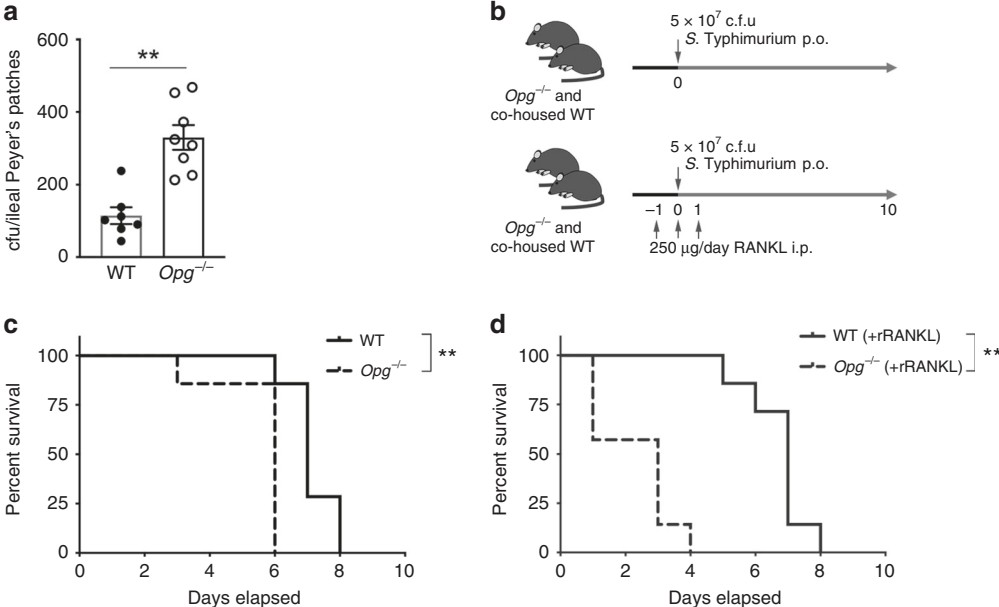

**Fig. 9 OPG-deficient mice have increased susceptibility to _S._ Typhimurium infection. a** $Opg^{-/-}$ or co-housed WT mice were orally infected with $5 \times 10^8$ c.f.u. of _S._ Typhimurium ($\Delta aroA$). Ileal Peyer's patches were collected 3 h after infection, and the numbers of colonies of _S._ Typhimurium ($\Delta aroA$) were counted. Data from two independent experiments are presented as the mean ± standard error. **$p < 0.01$ (Mann–Whitney $U$ test, $n = 8$ animals from two independent experiments). **b** Schematic cartoon depicting the experimental design of **c** and **d**. **c** $Opg^{-/-}$ or co-housed WT mice orally infected with $5 \times 10^7$ c.f.u. of _S._ Typhimurium were monitored for survival. **$p < 0.01$ (Kaplan–Meier test, $n = 7$ animals). **d** GST-RANKL was intraperitoneally injected into $Opg^{-/-}$ or co-housed WT mice for 3 days, starting 1 day before infection with $5 \times 10^7$ c.f.u. of _S._ Typhimurium. Mice were monitored for survival. **$p < 0.01$ (Kaplan–Meier test, $n = 7$ animals). The source data underlying panels **a**, **c** and **d** are provided as a Source Data file.

bacteria-specific high-affinity SIgA was abundantly produced in the cLP of $Opg^{-/-}$ mice under inflammatory conditions. Cecal patches are responsible for the generation of IgA$^+$ B cells that eventually migrate to the colonic lamina propria[40]. Considering that IgA$^+$ B cells were significantly increased in cecal patches of $Opg^{-/-}$ mice in conjunction with the expanded steady-state M-cell population, it is conceivable that these IgA$^+$ B cells migrate to the colonic lamina propria to reinforce commensal-specific IgA production under inflammatory conditions. These findings are consistent with the conventional view that M cells serve as entry sites for luminal antigens to facilitate mucosal SIgA responses in GALT[3]. However, recent reports have revealed that IgG responses to commensal bacteria are constitutively induced under physiological conditions[33,34]. In addition to SIgA, commensal-specific IgG circulating in the blood confronts the occasional intrusion of gut microbes into the internal milieu due to epithelial barrier dysfunction[33]. Peyer's patches and mesenteric lymph nodes have been recognized as the major inductive sites for the commensal-specific IgG response. However, our data suggested an important role for cecal patches, rather Peyer's patches, in the production of commensal-specific IgG3. A recent study revealed that colonic IgA-secreting cells were mainly generated in cecal patches[40]. In addition to IgA$^+$ B cells, we detected IgG$^+$ B cells in cecal patches. Because OPG deficiency increased the numbers of IgG$^+$ B cells in cecal patches, we presume that they are maintained by M-cell-dependent uptake of luminal antigens (i.e., commensal microbes). Consistent with this finding, $Opg^{-/-}$ mice displayed higher levels of commensal-specific IgG3 antibodies in the serum. Additionally, commensal-specific IgG1, which is only slightly induced in WT mice, was significantly elevated in $Opg^{-/-}$ mice. Our results support a potential contribution of M-cell-dependent antigen uptake to the production of commensal-specific IgG3.

The regulation of M-cell density by OPG seems to diminish antigen-specific mucosal and systemic immune responses mediated by IgA and IgG3, and thus may eventually create the

opportunity for systemic translocation of commensal microbes in association with epithelial barrier disruption under inflammatory conditions. Indeed, the activation of commensal-specific immune responses due to OPG deficiency resulted in the amelioration of disease symptoms, as well as splenomegaly, by thwarting bacterial translocation in DSS-induced colitis. A previous report demonstrated that luminal SIgA prevents the development of DSS-induced colitis[41]. Moreover, serum IgG plays a role in the neutralization of systemically translocated commensal bacteria and prevention of lethality in the same model[33]. Among IgG subclasses, IgG3 specifically recognizes commensal bacteria[34]. Indeed, our study demonstrated that adoptive transfer of IgG3$^+$ or IgA$^+$ B cells into $Rag1^{-/-}$ recipient mice ameliorated symptoms of colitis. Considering our data and the findings of previous reports, protection against DSS-induced colitis in the absence of OPG may result from the enhancement of commensal-specific IgA and IgG3 responses.

Although OPG is also expressed by several immune cell subsets, the bone marrow chimeric mice in this study demonstrated that deficiency in non-hematopoietic OPG is important for the alleviation of colitis. Despite this apparent disadvantage, M cells constitutively express OPG. There could be several reasons for this expression, which likely outweigh the risk of exacerbation of acute colitis. Our data demonstrate that M-cell expansion increased susceptibility to infection by M-cell-targeting pathogenic microbes, such as _S._ Typhimurium. An increase in M cells may also facilitate the translocation of botulinum toxins and the scrapie prion protein into the body[42,43]. Therefore, it is reasonable to consider that OPG limits mucosal invasion of pathogenic agents by regulating the number of M cells. Furthermore, excessive immune response in GALT may enhance the risk of autoimmune diseases, given that GALT could contribute to the differentiation of autoreactive T cells in an autoimmune arthritis model, as well as in a model of experimental autoimmune encephalomyelitis.

The biological significance of the RANKL/OPG balance has been primarily characterized in the context of bone homeostasis maintenance. Our study uncovered a novel link between the RANKL/OPG balance and self-regulation of M-cell density. Finetuning of M-cell differentiation is essential for the prevention of mucosal invasion, as well as the maintenance of immune homeostasis in both mucosal and systemic immune systems. Given that OPG also constitutes a biomarker for IBDs, namely, Crohn's disease and ulcerative colitis[18,19], RANKL/OPG imbalances might be involved in the pathogenesis of chronic inflammatory disorders and autoimmune diseases.

## Methods

**Animal experiments**. $Opg^{-/-}$ mice on a C57BL/6J background were generated using a standard gene targeting method[44]. C57BL/6J mice were purchased from CLEA Japan, Inc. (Tokyo, Japan) or Japan SLC, Inc. (Shizuoka, Japan). To rule out the influence of distinct commensal bacteria, after weaning, all mice were housed in the same cage for at least 3 weeks. These mice were maintained under conventional conditions at the Graduate School of Medicine, Hokkaido University or the Institute of Medical Science (Sapporo, Japan), The University of Tokyo (IMSUT, Tokyo, Japan). All mice were between 6 and 10 weeks of age at the onset of experiments.

**Ethical statement for animal research**. Protocols approved by Hokkaido University, IMSUT, and Keio University were used for all animal experiments.

**Digital RNA-seq and data analyses**. For the separation of single epithelial cells, Peyer's patches were collected and incubated in Hank's balanced salt solution (HBSS; Life Technologies, Grand Island, NY, USA) containing 1 mM dithiothreitol (DTT) and 10 µM Y27632 (Wako Pure Chemical Industries, Osaka, Japan), followed by an additional incubation at 37 °C with shaking in HBSS containing 20 mM ethylenediaminetetraacetic acid (EDTA), 10 µM Y27632, and 12.5 mM 4-(2-hydroxyethyl)-1-piperazineethanesulfonic acid (HEPES). Epithelial cells were separated using TrypLE Select (Gibco, Grand Island, NY, USA) with 10 µM Y27632. Using a BD FACSAria™ III (BD Biosciences, San Jose, CA, USA), GP2⁺ and GP2⁻ epithelial cells from the CD45⁻EpCAM⁺CD24⁻ fraction of the Peyer's patch epithelial layer in C57BL/6J mice were sorted. Apoptotic and necrotic cells were excluded by co-staining with eFluor450-labeled recombinant Annexin V and 7-AAD, respectively. Libraries for digital RNA sequencing[20,45] were prepared from 100 cells, as described previously[46]. Three independent libraries each were generated from M cells and epithelial cells with different sample indexes. All six libraries were sequenced together in a single MiSeq run (150 cycles; Illumina kit, Illumina, San Diego, CA, USA), which was performed twice to obtain additional reads. Sequencing data were mapped against the mouse genome (mm10 assembly) using TopHat2. Differential gene expression among the digital RNA-seq data was determined using DESeq2. Enrichment analysis was based on the KEGG functional hierarchy. $P$ values for the enrichment test were calculated by the GAGE algorithm[47], and the false-discovery rate was calculated from the $p$ value for multiple testing. The enrichment results were visualized using FuncTree[48] and modified with Adobe Illustrator (Adobe Systems Inc., San Jose, CA, USA). All sequencing data sets have been deposited in the Genome Expression Omnibus database under accession number GSE108529.

**Preparation of FAE and conventional epithelial monolayers**. Peyer's patches from the ileum and cecal patches were soaked in ice-cold HBSS containing 30 mM EDTA and 5 mM DTT. After incubation with gentle shaking for 20 min on ice, FAE monolayers containing both dome and crypt regions were carefully separated from lymphoid follicles by manipulation with a fine needle under stereomicroscopic monitoring[13,14]. Conventional epithelial monolayers were isolated from small pieces of the ileum and cecum in the same manner.

**Immunofluorescence staining**. For whole-mount staining, each isolated epithelium was fixed with 4% paraformaldehyde (PFA) in phosphate-buffered saline (PBS) for 30 min. For conventional observation, frozen sections (10 µm in thickness) were mounted on poly-L-lysine-coated glass slides, air-dried, and fixed with 4% PFA in PBS for 30 min. After preincubation with 10% normal donkey serum for 1 h, specimens were incubated with primary antibodies diluted in Can Get Signal® Immunostain solution B (Toyobo, Osaka, Japan) overnight at 4 °C with gentle shaking, followed by incubation with appropriate secondary antibodies for 2 h at room temperature. For detection of filamentous actin, fluorescently labeled phalloidin (Acti-Stain 670 diluted at 1:400; Cytoskeleton, Inc., Denver, CO, USA) was included during the incubation with the primary antibodies. For nuclear staining, we applied 4′,6-diamidino-2-phenylindole, dihydrochloride (DAPI), TOPRO-3, or Hoechst 33342 (Life Technologies) after sample incubation with the secondary antibodies. Specimens were observed using a confocal laser microscope

FV300 or FV1000 (Olympus, Tokyo, Japan) after mounting with SlowFade Gold antifade reagent (Life Technologies).

For immunoelectron microscopy, PFA-fixed Peyer's patches were processed for use by the pre-embedding silver-intensified immunogold method. Frozen sections were incubated with anti-OPG (R&D Systems, Minneapolis, MN, USA) and subsequently reacted with donkey anti-goat IgG covalently linked to 1.4-nm gold particles (BB International, Cardiff, UK). After silver enhancement (HQ silver; Nanoprobes, Yaphank, NY, USA), sections were osmicated, dehydrated, and embedded in Quetol 812 (Nisshin EM, Tokyo, Japan). Ultrathin sections were prepared and stained with uranyl acetate and lead citrate for observation under a transmission electron microscope (H-7100, Hitachi, Tokyo, Japan). The primary antibodies used for immunofluorescence staining are as follows: anti-OPG antibody (catalog number AF549; R&D systems, 1:400); anti-GP2 antibody (D278-3; MBL, 1:400); anti-Tnfaip2 antibody (Kimura et al.[14]: 200); anti-SpiB antibody (AF7204; R&D Systems, 1:200); anti-RelB antibody (sc-226; Santa Cruz Biotechnology, 1:100); and anti-RANKL antibody (14-5952; eBioscience, 1:100).

**Fluorescent in situ hybridization**. FISH was performed using the QuantiGene View RNA ISH Cell Assay (Affymetrix Inc., Santa Clara, CA, USA) with slight modifications in the fixation and protease digestion steps. Briefly, frozen sections (10 µm in thickness) were mounted on poly-L-lysine-coated glass slides, air-dried, and fixed with 4% PFA in PBS for 30 min. After three washes with 50 mM glycine in PBS, the specimens were pretreated with a detergent solution (Affymetrix) for 10 min. In the current study, proteinase K digestion was omitted. The remaining processes in the protocol were performed in accordance with the manufacturer's instructions. Specific oligonucleotide probe sets against *Spib* (VB1-13735) and *Tnfrsf11b* for *Opg* mRNA (VB6-13966) were purchased from Affymetrix.

**Ligated intestinal loop assay**. To prepare ligated intestinal loops, the mice were anesthetized using an isoflurane vaporizer. Aliquots of $1 \times 10^{11}$/ml fluorescent polystyrene latex beads (20-nm diameter; Life Technologies) were injected into the ligated intestinal loop as described previously[14]. The mice were euthanized 120 min after the injection and subjected to immunofluorescence experiments.

**Scanning electron microscope observations**. Three C57BL/6J and three $Opg^{-/-}$ mice (10–12 weeks old) were used in these experiments. After an overnight fasting, the animals were anesthetized with sodium pentobarbital (60 mg/kg body weight), and subsequently laparotomized for construction of a ligated ileal loop. Each intestinal loop measured approximately 15 cm in length and contained two Peyer's patches in a middle portion; a polyethylene tube approximately 2-mm-thick was inserted prior to closure of the loop. The intestinal lumen was gently washed with warmed saline through the tube. The animals were then fixed by transcardial perfusion with a mixture of 1.25% glutaraldehyde and 1% paraformaldehyde in 0.1 M phosphate buffer, pH 7.3. During fixation, the ileal loop was inflated with the same fixative through the lavage tube. The distended Peyer's patches were dissected from the ileal loops with razor blade, postfixed in 1% OsO₄ for 2 h, and conductive-stained in 1% tannic acid, and subsequently in 1% OsO₄. Osmicated specimens were dehydrated with ethanol and critical-point-dried with liquid CO₂. Dried specimens were coated with osmium in a plasma osmium coater (Nippon Laser and Electronics Laboratory, Nagoya, Japan) and examined under a Hitachi SU8010 scanning electron microscope (Hitachi).

**RANKL administration**. The primers 5′-CACCCCCGGGCAGCGCTTCTCAGGA GCT-3′ and 5′-GAGACTCGAGTCAGTCTATGTCCTGAAC-3′ (Sigma Genosys Inc., Woodlands, Texas, USA) were used for a polymerase chain reaction (PCR) to amplify a cDNA clone of RANKL. The PCR fragment was subcloned into the pGEX-4T-2 vector (GE Healthcare, Waukesha, WI, USA) after digestion by *Sma*I and *Xho*I. The construct was transformed into the BL21 *Escherichia coli* strain for glutathione-*S*-transferase (GST) fusion protein expression. The culture was induced with 0.1 mM isopropyl β-D-1-thiogalactopyranoside for 16 h at 20 °C, and the GST-RANKL was purified from bacterial lysate by affinity chromatography on a Glutathione-Sepharose 4B (GE Healthcare) followed by dialysis against multiple changes of PBS. Recombinant GST used as a control was prepared by the same method using an empty pGEX-4T-2 vector. Purified protein was administered to mice by intraperitoneal injections of 250 µg per day for up to 4 days[14]. Mice were sacrificed 24 h after the final administration, and samples from these mice were subjected to various assays, as described below.

**Quantitative reverse transcriptase PCR**. Total RNA from epithelium samples isolated from mice injected with GST or GST-RANKL were prepared using TRIzol (Life Technologies). First-strand cDNA synthesis was completed using ReverTra Ace (Toyobo). Quantitative PCR reactions were conducted in Rotor Gene 6000 equipment (Qiagen, Hilden, Germany) using a KAPA SYBR Green Fast PCR kit (KAPA Biosystems, Woburn, MA, USA). The specific primers used in these assays are shown in Supplementary Table 1.

**Western blotting**. Small pieces of isolated epithelial sheets (approximately 30 mm in length) from the intestines of GST-RANKL-injected mice were washed with ice-

cold HBSS and suspended in 0.5 ml of lysis buffer [10 mM HEPES-NaOH pH 7.9, 1.5 mM $MgCl_2$, 10 mM KCl, 0.1 mM EDTA, 1 mM DTT, 0.1% NP-40, and protease inhibitor cocktails (Roche, Mannheim, Germany)]. The epithelial sheets were kept on ice for 15 min and then vortexed vigorously for 15 s. These homogenates were centrifuged at $850 \times g$ for 10 min, and 6× SDS sample buffer was added to each of the resulting supernatants (post-nuclear extract), after which they were stored at −30 °C. The remaining pellet was washed with 0.5 ml of lysis buffer, centrifuged under the same conditions, resuspended in 50 μl of ice-cold nuclear extraction buffer [20 mM HEPES-NaOH pH 7.9, 1.5 mM $MgCl_2$, 400 mM NaCl, 0.1 mM EDTA, 0.1% NP-40, 10% glycerol, 10 mM DTT, and protease inhibitor cocktail (Roche)], and incubated on ice for 30 min with intermittent mixing. The samples were then centrifuged at $14,000 \times g$ for 30 min, and 6× SDS sample buffer was added to each of the resulting supernatants (nuclear extracts), after which they were stored at −30 °C. Equal concentrations of proteins from post-nuclear and nuclear extracts were separated on 12% polyacrylamide gels and transferred to PVDF membranes. Standard immunostaining was performed using the ECL enhanced chemiluminescence technique (GE Healthcare, Chicago, IL, USA). The primary antibodies used for western blotting are as follows: anti-RelB antibody (catalog number: sc-226; Santa Cruz Biotechnology, dilution 1:2000); anti-p100/p52 antibody (#4882; Cell Signaling Technology, 1:1000); anti-Lamin A/C antibody (GTX1116777; GeneTex, 1:2000); and anti-Rpt4 antibody (BML-PW8830; Enzo Life Science, 1:2000). The chemiluminescence and bright field images of protein blotted membranes were acquired with a high-performance intensified CCD camera (LAS4000, FUJI film). The uncropped and unprocessed scans of all blots are provided in a Source Data file.

**Image cytometry analysis**. Confocal images were acquired in the photon counting mode of a FV1000 confocal microscopy system with FV10-ASW software (Olympus). To calculate the population of OPG-expressing cells, we selected the region of M cells by setting a threshold of signal intensities for OPG and/or Spi-B and then calculated mean fluorescence intensities within these regions by using ImageJ software. Obtained values were plotted by Microsoft Excel software or saved as converted from a comma-separated values (csv) format to a flow cytometry standard (fcs) format using a CsvToFcs module of GenePattern software [https://gp.indiana.edu/gp/pages/index.jsf]. The fcs files were opened and analyzed by FlowJo software (TreeStar Inc., Ashland, OR, USA).

**DSS colitis**. We performed experiments with our DSS-induced colitis model in two different facilities, Hokkaido University and IMSUT. $Opg^{-/-}$ mice and co-housed WT mice or littermate $Opg^{+/-}$ mice consumed DSS (dextran sulfate sodium salt; MP Biomedicals LLC., Santa Ana, CA, USA) in their drinking water for up to 8 days. On the basis of the results of preliminary experiments undertaken in each facility to estimate the optimal DSS concentration, we set the DSS concentration at 2.5% for Hokkaido University and at 1.5% for IMSUT. Body weight and fecal clinical score were measured every day during the experimental period for DSS colitis. Feces were assessed by diarrhea score as follows: 0, normal stool consistency with negative ColoScreenES; 1, soft stools with positive ColoScreenES; 2, very soft stools with traces of blood; and 3, watery stools with visible rectal bleeding. The presence of occult blood was tested by ColoScreen (Helena Laboratories, Beaumont, TX, USA). We obtained similar results from the two different facilities using these experimental conditions.

**Flow cytometry**. Colons, distal Peyer's patches (ileal Peyer's patches), and cecal patches were dissociated with RPMI 1640 medium containing 12.5 mM HEPES (pH 7.4), 2% fetal bovine serum, 0.5 mg/ml collagenase (Wako Pure Chemical Industries), and 0.5 mg/ml DNase I (Wako Pure Chemical Industries) at 37 °C for 30 min[49]. Cells were isolated by Percoll gradient separation.

Isolated cells were incubated with anti-CD16/CD32 (FcγR) antibody (catalog number 101319, BioLegend, dilution 1:500) to block non-specific reactions and subsequently stained by using specific antibodies. For intracellular cytokine staining, isolated cells were cultured with phorbol 12-myristate 13-acetate, ionomycin, and brefeldin A for 4.5 h prior to cell surface staining. After surface staining, dead cells were stained by Live/Dead Aqua. Cells were fixed by BD Cytofix/CytoPerm (BD Biosciences) for 15 min. After fixation, cells were stained by various antibodies directed against intracellular cytokines. Flow cytometric analyses were performed using an LSRII (BD Biosciences).

The antibodies used for flow cytometry are as follows: redFluor710 anti-mouse CD3ε antibody (catalog number 80-0032, Tonbo Biosciences, dilution 1:800); eFluor450 anti-mouse CD3ε antibody (48-0031-80, Thermo Fisher Scientific, 1:800); APC-H7 anti-mouse CD4 antibody (560246, BD Bioscience, 1:800); PE anti-mouse CD8a antibody (12-0081-81, Thermo Fisher Scientific, 1:800); PerCP-Cyanine5.5 anti-CD11b (45-0112-80, Thermo Fisher Scientific, 1:800); PE-Cy7 anti-mouse CD11c antibody (117318, BioLegend, 1:800); APC anti-mouse CD24 antibody (101814, BioLegend, 1:400); PE anti-mouse CD45 antibody (103105, BioLegend, 1:800); FITC anti-mouse CD45.1 antibody (35-0453, Tonbo Bioscience, 1:800); APC-eFluor 780 anti-mouse CD45.2 antibody (47-0454-82, Thermo Fisher Scientific, 1:800); APC-Cy7 anti-mouse CD45R (552094, BD Bioscience, 1:800); FITC anti-mouse CD95 antibody (561979, BD Bioscience, 1:400); APC anti-mouse CD103 antibody (121414, BioLegend, 1:400); PE anti-mouse CD326 antibody

(118206, BioLegend, 1:400); eFluor 660 anti-GL7 antibody (50-5902-80, Thermo Fisher Scientific, 1:400); Alexa488 anti-GP2 antibody (D278-A48, MBL, 1:800); Alexa488 anti-FoxP3 antibody (53-5773-80, Thermo Fisher Scientific, 1:100); eFluor 660 anti-Gata3 antibody (50-9966-41, Thermo Fisher Scientific, 1:100); Alexa488 anti-mouse Ly6G/Ly6C antibody (108419, BioLegend, 1:400); APC anti-Ly6C (560595, BD Bioscience, 1:400); APC-Cy7 anti-mouse Ly6G antibody (560600, BD Bioscience, 1:400); biotin anti-IgA antibody (556978, BD Bioscience, 1:800); FITC anti-IgG antibody (406001, BioLegend, 1:800); PerCP-Cy5.5 anti-IFN-γ antibody (45-7311-80, Thermo Fisher Scientific, 1:400); PE anti-IL-17A antibody (506903, BioLegend, 1:400); eFluor450 anti-MHC class II antibody (48-5321-80, Thermo Fisher Scientific, 1:800); PE anti-ROR-γt antibody (562607, BD Bioscience, 1:100); and PerCP-Cy5.5 anti-T-bet antibody (45-5825-82, Thermo Fisher Scientific, 1:100). Gating strategies are included in Supplementary Fig. 15.

**Bug-Ig flow cytometry**. We used a modified version of the Bug-Ig flow cytometry protocol[33,34]. Bacteria were washed with sterile PBS and resuspended with sterile PBS containing 1% BSA (1% BSA/PBS). Bacteria were mixed with or without serum diluted 1:25 in 1% BSA/PBS. After they had been washed with 1% BSA/PBS, bacteria were stained using biotin anti-IgA, PE/Dazzle594 anti-mouse IgD (405742, BioLegend, 1:200), PE anti-mouse IgM (562033, BD Bioscience, 1:200), PE/Cy7 anti-IgG1 antibody (406613, BioLegend, 1:200), PerCP/Cy5.5 anti-IgG2a (407111, BioLegend, 1:200), PE anti-mouse IgG2b (406708, BioLegend, 1:200), FITC anti-mouse IgG3 (553403, BD Bioscience, 1:800), or streptavidin-APC. Bacteria were resuspended in 1% BSA/PBS with DAPI (Life Technologies, Grand Island, NY, USA) and analyzed using a FACSAria (BD Biosciences).

**Bone marrow chimeric experiment**. Bone marrow cells ($1 \times 10^7$) were intravenously transferred into sub-lethally-irradiated recipient 8-week-old mice. Chimeras were additionally maintained for 6–8 weeks under CV conditions. The mice were treated with 0.5 mg/ml ampicillin and 1 mg/ml neomycin in drinking water to prevent inflammation for 2 weeks after the transfer. The efficiency of transplantation was confirmed by flow cytometric analysis of white blood cells based on CD45.1 and CD45.2 expression at 6 or 8 weeks post-transfer; mice whose blood lymphocytes were dominantly occupied by donor-derived hematopoietic cells (approximately 70–90%) were used for studies.

**Immunization and TT-specific ELISA**. rSalmonella-ToxC (ΔaroA, ΔaroD) and TT were kindly provided by the BIKEN Foundation (Osaka, Japan). $Opg^{-/-}$ mice and co-housed WT mice were orally or intraperitoneally infected with $5 \times 10^7$ or $5 \times 10^5$ colony-forming units (c.f.u.) of rSalmonella-ToxC[1,25]. TT-specific IgA and IgG in feces and serum were measured by ELISA. Flat-bottomed, 96-well Maxisorp Nunc-immuno plates (Thermo Fisher Scientific, Waltham, MA, USA) were coated overnight with 500 ng/well of TT. Plates were blocked with 2% BSA in PBS, and optically diluted fecal extracts and sera were added into the plate wells. The Mouse IgA ELISA Quantitation Set (Bethyl Laboratories, Inc., Montgomery, TX, USA.) and Mouse IgG ELISA Quantitation Set (Bethyl Laboratories, Inc.) were used for antibody detection. To produce horse radish peroxidase signals, a 1-Step Ultra TMB-ELISA (Thermo Fisher Scientific) was used.

**Colony-forming unit measurement after oral infection**. Tetracycline-resistant S. Typhimurium (ΔaroA) and streptomycin-resistant S. Typhimurium (ΔorgA) were kindly provided by Dr. H. Matsui (Kitasato University) and Dr. Y. Kim (Keio University), respectively. S. Typhimurium (ΔorgA) is a strain that lacks type III secretion system (T3SS) needle components in SPI-1 pathogenicity island[50], and this strain was described as a SPI-1 mutant in this study. $Opg^{-/-}$ mice and co-housed WT mice were infected with $5 \times 10^8$ c.f.u. of S. Typhimurium (ΔaroA) or $5 \times 10^7$ c.f.u. of SPI-1 mutant. The distal three Peyer's patches (ileal Peyer's patches) and mesenteric lymph nodes were collected at 3 or 48 h post-infection of S. Typhimurium (ΔaroA) and S. Typhimurium (SPI-1 mutant). Peyer's patches were incubated in PBS containing 500 μg/ml gentamicin (Thermo Fisher Scientific) for 30 min and washed with PBS containing 1 mM DTT. The specimens were homogenized in sterile PBS and subsequently centrifuged briefly to roughly remove host cells. Ten-fold serial dilutions of the resulting supernatants were plated on Luria-Bertani agar plates containing 15 μg/ml tetracycline (Tokyo Chemical Industry Co. Ltd, Tokyo, Japan) or 50 μg/ml streptomycin (Wako Pure Chemical Industries) to determine the number of c.f.u.[51].

**Salmonella infection survival rate**. $Opg^{-/-}$ mice and co-housed WT mice were infected with $5 \times 10^7$ c.f.u. of S. Typhimurium χ3306. GST-RANKL (10 mg/kg) was administered intraperitoneally once daily for 3 days, beginning from 1 day prior to the onset of infection.

**Statistics**. Differences between mean values for two or more groups were analyzed by Student's t-test or one-way ANOVA with Tukey's test, respectively, using Prism (GraphPad Software, La Jolla, CA, USA). When variances were not homogeneous following an F-test, the values were statistically analyzed by Welch's t-test. When values followed a normal distribution according to the results of the Shapiro–Wilk

test, the nonparametric Mann–Whitney $U$ test was performed. For analysis of the survival rate, the Kaplan–Meier test was performed.

**Reporting summary**. Further information on research design is available in the Nature Research Reporting Summary linked to this article.

## Data availability

Digital RNA sequencing data sets have been deposited in the Genome Expression Omnibus database under the accession number GSE108529. Metagenomic 16S rRNA sequencing data have been deposited in the DNA Data Bank Japan (DDBJ) under the accession number "DRP004759". The source data underlying Figs. 1c, g, 2b, c, e, f, 3b, c, 4b-d, 5a-g, 6a, b, d-f, 7a-h, 8b, c, e, f, 9a, c, and 9d and Supplementary Figs. 5b, 6, 7a–e, 8, 9a–d, 10a, b, 11a–d, and 12 are provided as a Source Data file. The dataset generated and analyzed during the current study are available from the corresponding authors upon reasonable request.

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

## Acknowledgements

We thank Prof. H. Ohno and Dr. T. Kanaya for valuable suggestions, and Ms. K. Fukuhara for help in performing digital RNA sequencing. This study was supported by JSPS or MEXT Grants-in-Aid for Scientific Research 25460261, 16K08457 (to S.K.), 18H06033 (to Y.N), 16H01369, 17H04089, 18H04680, 25293114, 26116709 (to K.H.), AMED-Crest 16gm0000000h0101, 17gm1010004h0102, 18gm1010004h0103, and 19gm1010004s0104 (to K.H.), and JST PRESTO JPMJPR19H3 (to S.K.). This study was also supported by grants from the Research Foundation for Opto-Science and Technology, the Mochida Memorial Foundation for Medical and Pharmaceutical Research, The Food Science Institute Foundation, the SENSHIN Medical Research Foundation (to S.K.), the Takeda Science Foundation (to S.K. and K.H.), the NOVARTIS Foundation JAPAN for Promotion of Science (to K.H.), and the Grant for Joint Research Project of the Institute of Medical Science, the University of Tokyo (to K.H.).

## Author contributions

Conceptualization: S.K. and K.H.; methodology, S.K., Y.N., N.K., E.K., and K.S.; investigation: S.K., Y.N., N.K., E.K., K.S., M.M., H.T.I., M.H., T.Y., T.I., and K.H.; resources: M.N., N.U., S.S., and T.K.; writing—original draft, S.K. and Y.N.; writing—review & editing, S.K., T.I., and K.H.; supervision, S.K. and K.H.; funding acquisition, S.K. and K.H.

## Competing interests

The authors declare no competing interests.
