## [Peer Review File · Nature Communications]

Editorial Note: This manuscript has been previously reviewed at another journal that is not operating a transparent peer review scheme. This document only contains reviewer comments and rebuttal letters for versions considered at Nature Communications. Parts of this peer review file have been redacted as indicated to remove third-party material where no permission to publish could be obtained.

Reviewers' comments:

Reviewer #1 (Remarks to the Author):

The authors have addressed many important concerns raised by the reviewers. The manuscript convincingly describes a function of OPG in regulating M cell differentiation and intestinal immunity.

Reviewer #2 (Remarks to the Author):

Thanks to the authors for their thoughtful response to the reviewers' comments, as well as the insightful comments of the other referees. I agree with the responses and modifications to the manuscript.

Reviewer #3 (Remarks to the Author):

In their manuscript titled 'Osteoprotegerin-dependent M-cell self-regulation balances gut infection and immunity', Kimura and colleagues elegantly show that osteoprotegerin (OPG) limits differentiation of intestinal stem cells into M cells and that in its absence more M cells appear in the follicle associated epithelium (FAE). By combining *in vivo* experiments in wildtype and *Opg*^{-/-} mice with *in vitro* intestinal organoids, the authors clearly demonstrate that *Opg* limits Rankl signalling, as it does in bone, and thus limits M-cell differentiation in the intestine. Furthermore, the authors claim that an increase in M cells due to lack of *Opg* enhances the production of antibodies against commensal bacteria, thereby limiting SDS induced colitis in mice while on the other hand mice become more susceptible for infections with M cell targeting microbes such as Salmonella.

In general, the authors have answered to the issues raised by me and produced a well written revised manuscript. A number of issues still remain I feel are in need of addressing.

- Authors response 3-6: I think adding Reviewer Figure 8 to the manuscript can be of value.
- Authors response 3-13 and Fig. S10: the conclusion drawn by the authors based on these experiments (OPG deficiency protects from colitis because IgG3 and IgA titers are induced in *Opg*^{-/-} animals and B cells producing these isotypes protect *Rag1*^{-/-} mice from colitis) is circumstantial at best. To make this claim stronger, the authors should repeat the adoptive transfer experiments with colonic B cells (supposedly commensal-specific) from *Opg*^{-/-} and WT mice.
- Authors response 3-15: considering the lack of consensus on this specific matter, I suggest to omit the data on *Spib*^{-/-} mice.
- Fig. 5G (also in response to the authors rebuttal) : based on these data it is not possible to say that the systemic immune response is not altered in *Opg*^{-/-} mice. Whereas the effect might not be big, purely based on these experiments there does seem to be an effect on antibody production. As the authors did not perform experiments to assess the recall response to immunization, which is also an important readout when assessing an adaptive immune response, it is impossible to state that there is no immunological effect seen in *Opg*^{-/-} mice. While this does not disqualify the rest of the data presented here, the authors should add a statement addressing this to the manuscript.

- Figure 4: both *SpiB* and *TNFAIP2* seem to go up in the same way in the cecal epithelium as in the ileal villous epithelium upon Rankl injection. I believe this contradicts the notion that the cecal

epithelium is unresponsive to Rankl (line 213-214). Based on these results, the cecal epithelium does respond to Rankl stimulation in WT animals, but GP2 is not upregulated. The phrasing should be altered.

- Fig. 9: administration of RANKL does not seem to influence survival of WT mice, while it dramatically decreases survival of Opg^{-/-} mice. Can the authors explain this?

Minor points

- The description of the panels in the legend in figure 3 does not correlate to the figure.

- A description of the image cytometry analysis performed by the authors is lacking in the materials & methods section.

Reviewer #1 (Remarks to the Author):

The authors have addressed many important concerns raised by the reviewers. The manuscript convincingly describes a function of OPG in regulating M cell differentiation and intestinal immunity.

Response: We appreciate reviewer#1 for the positive evaluation of our revised manuscript.

Reviewer #2 (Remarks to the Author):

Thanks to the authors for their thoughtful response to the reviewers' comments, as well as the insightful comments of the other referees. I agree with the responses and modifications to the manuscript.

Response: We appreciate reviewer#2 for the positive comment on our revised manuscript.

Reviewer #3 (Remarks to the Author):

In their manuscript titled 'Osteoprotegerin-dependent M-cell self-regulation balances gut infection and immunity', Kimura and colleagues elegantly show that osteoprotegerin (OPG) limits differentiation of intestinal stem cells into M cells and that in its absence more M cells appear in the follicle associated epithelium (FAE). By combining in vivo experiments in wildtype and Opg^{-/-} mice with in vitro intestinal organoids, the authors clearly demonstrate that Opg limits Rankl signaling, as it does in bone, and thus limits M-cell differentiation in the intestine. Furthermore, the authors claim that an increase in M cells due to lack of Opg enhances the production of antibodies against commensal bacteria, thereby limiting SDS induced colitis in mice while on the other hand mice become more susceptible for infections with M cell targeting microbes such as Salmonella.

In general, the authors have answered to the issues raised by me and produced a well written revised manuscript. A number of issues still remain I feel are in need of addressing.

- Authors response 3-6: I think adding Reviewer Figure 8 to the manuscript can be of value.

Response: We appreciate the reviewer#3 for finding significant progress in our revised manuscript and making valuable comments. Following the reviewer's suggestion, we added Reviewer Figure 8 to the manuscript as Fig. S6 and mentioned it in line 214-215 as follows:

The expression level of RANK mRNA in the cecal epithelium was similar to that in the ileal epithelium (Fig. S6).

- Authors response 3-13 and Fig. S10: the conclusion drawn by the authors based on these experiments (OPG deficiency protects from colitis because IgG3 and IgA titers are induced in Opg^{-/-} animals and B cells producing these isotypes protect Rag1^{-/-} mice from colitis) is circumstantial at best. To make this claim stronger, the authors should repeat the adoptive transfer experiments with colonic B cells (supposedly commensal-specific) from Opg^{-/-} and WT mice.

Response:

Following the reviewer's suggestion, we performed the adoptive transfer of colonic B cell subsets from Opg^{-/-} and WT mice into Rag1^{-/-} recipient mice. Consequently, the adoptive transfer of IgG3⁺ and IgA⁺ cells from both groups exerted a protective effect on the DSS colitis model, as evidenced by the improvement of fecal clinical scores, increasing colon weight, and shortening colon length (**Reviewer Figure A**). However, there is no significant difference in the protective effect of the B cell subsets between WT and Opg^{-/-} mice, suggesting that M-cell hyperplasia may be required to maintain the activated B cell status. Given that the recipient Rag1^{-/-} recipient mice possess just an ordinary number of M cells, we infer that the B cell subsets from Opg^{-/-} mice may be gradually inactivated or normalized after the transfer. M-cell hyperplasia due to OPG deficiency should constitutively promote IgA- as well as IgG3-class switching in the gut-associated lymphoid tissue, and thus the continuous supply of IgA⁺ and IgG3⁺ B cells may be essential to suppress gut inflammation potently.

Figure A. Adoptive transfer of IgG3⁺ and IgA⁺ cells from WT and *Opg*^{-/-} mice showed a similar protective effect on the development of DSS-induced colitis.

IgA⁺ and IgG3⁺ cells were collected from WT and *Opg*^{-/-} mice. These cells were intravenously transferred into *Rag1*^{-/-} mice. After 3 days, these mice were administered with 2% dextran sodium sulfate (DSS) in drinking water for 7 days. (A) Daily changes in body weight during DSS-induced colitis. (B) Stool scores were measured as described in the Method section. (C, D) Colon length and colon thickening were measured on day 8. Data from two independent experiments are shown as the mean ± standard deviation. ***p*<0.01, **p*< 0.05 (one-way ANOVA-test, *n* = 5-6).

- Authors response 3-15: considering the lack of consensus on this specific matter, I suggest to omit the data on Spib^{-/-} mice.

Response: Following the reviewer's suggestion, we omitted the data and descriptions of *Spib*^{-/-} mice.

- Fig. 5G (also in response to the authors rebuttal) : based on these data it is not possible to say that the systemic immune response is not altered in Opg^{-/-} mice. Whereas the effect might not be big, purely based on these experiments there does seem to be an effect on antibody production. As the authors did not perform experiments to assess the recall response to immunization, which is also an important readout when assessing an adaptive immune response, it is impossible to state that there is no immunological effect seen in Opg^{-/-} mice. While this does not disqualify the rest of the data presented here, the authors should add a statement addressing this to the manuscript.

Response: We thank the reviewer for raising the significant issue. As the reviewer pointed out, OPG deficiency slightly but significantly attenuated antigen-specific IgG production after systemic immunization with the vaccine strain of *Salmonella*. We agree that OPG, most likely expressed by immune cell subset, may play a role in the induction of systemic immune response, although further experiments using conditional knockout mice is necessary to verify this notion. Following the reviewer's suggestion, we added the cellular distribution of OPG as Supplemental Figure S8. We made an additional statement in the Results (line 268-272) as follows: “These results indicated that the absence of *Opg* consolidates the IgA and IgG responses to luminal antigens. In contrast, the IgG response to systemic immunization was slightly, but significantly, lower in *Opg*^{-/-} mice (Fig. 5G). This result implies that OPG, most likely expressed by certain immune cell subsets (Fig. S8), may play a role in the induction of systemic immune response.”

- Figure 4: both *SpiB* and *TNFAIP2* seem to go up in the same way in the cecal epithelium as in the ileal villous epithelium upon *Rankl* injection. I believe this contradicts the notion that the cecal epithelium is unresponsive to *Rankl* (line 213-214). Based on these results, the cecal epithelium does respond to *Rankl* stimulation in WT animals, but GP2 is not upregulated. The phrasing should be altered.

Response: Following the reviewer's suggestion, we amend this sentence as below (Line 220-223; underlined).

“Quantitative PCR analysis also confirmed a remarkable (approximately 2×10^4 -fold) upregulation of *Gp2* in the cecal epithelium of *Opg*^{-/-} mice upon treatment with RANKL, whereas the *Gp2* expression level in this region of WT mice was unchanged by RANKL treatment (Fig. 4B). Nevertheless, the expression of *SpiB* and *Tnfaip2*, the early-to-middle M-cell differentiation makers, were upregulated even in the cecal epithelium of WT mice upon RANKL injection. Thus, OPG seems to suppress RANKL-RANK signaling mainly at the late stage of M-cell differentiation.”

- Fig. 9: administration of RANKL does not seem to influence survival of WT mice, while it dramatically decreases survival of *Opg*^{-/-} mice. Can the authors explain this?

Response: We thank the reviewer for pointing out another significant issue. In this experiment, mice were infected with wild-type *Salmonella enterica* serovar Typhimurium (*S. Typhimurium*) just after the 2nd administration of RANKL (Day 0). We previously showed that *S. Typhimurium* gains entry into the body via a GP2-dependent pathway (Hase et al., *Nature* 2009, 462:226-30). At Day 0 (24 h after the first RANKL administration), GP2 expression was induced ~~elevated~~ in the intestinal villi of

Opg^{-/-} mice but not in WT mice (Figure B). This result is consistent with our previous reports that, in wild-type mice, GP2 is not induced until two days after the RANKL administration (Kanaya et al., *Nature Immunol* 2012, 13:729-36; Kimura et al., *Mucosal Immunol.*, 2015, 8:650-60). Thus, the influence of RANKL administration on the *Salmonella* infection was considered to be minimal in WT mice at 24 h after the onset of RANKL treatment.

Figure B Rapid induction of *Gp2* in *Opg*^{-/-} mice by RANKL administration

The expression levels of M-cell associated genes—*Gp2*, *Spib*, *Tnfrsf25*—were measured by quantitative PCR. mRNA was collected on the second day of the RANKL administration that was the same as salmonella infection (day0). Results were represented as relative to *Gapdh* expression. Data are the mean ± standard deviation (*n* = 3).

[Redacted]

Figure C. Time-course analysis of *Gp2* expression in intestinal villous epithelium of wild-type mice after intraperitoneal injection with RANKL. The Graph was transferred from Figure 1a in Kanaya et al., *Nature Immunol* 2012, 13:729-36).

Minor points

- The description of the panels in the legend in figure 3 does not correlate to the figure.

Response: We corrected the figure legend.

- A description of the image cytometry analysis performed by the authors is lacking in the materials & methods section.

Response: We added a description of the image cytometry analysis in the materials and methods section (line 481-490).

REVIEWERS' COMMENTS:

Reviewer #3 (Remarks to the Author):

The authors have addressed all the issues raised by me appropriately. I agree with the answers provided by the authors and the changes made to the manuscript. The revised manuscript provides a nice insight into the function of OPG in mucosal immunology.